# Cryo-EM structures of ρ1 GABA$_A$ receptors with antagonist and agonist drugs

Chen Fan [1,2,3] ✉, John Cowgill [2], Rebecca J. Howard [1,2] ✉ & Erik Lindahl [1,2] ✉

The family of ρ-type GABA$_A$ receptors includes potential therapeutic targets in several neurological conditions, and features distinctive pharmacology compared to other subtypes. Here we report four cryo-EM structures with previously unresolved ligands, electrophysiology recordings, and molecular dynamics simulations to characterize binding and conformational impact of the drugs THIP (a non-opioid analgesic), CGP36742 (a phosphinic acid) and GABOB (an anticonvulsant) on a human ρ1 GABA$_A$ receptor. A distinctive binding pose of THIP in ρ1 versus α4β3δ GABA$_A$ receptors offers a rationale for its inverse effects on these subtypes. CGP36742 binding is similar to the canonical ρ-type inhibitor TPMPA, supporting a shared mechanism of action among phosphinic acids. Binding of GABOB is similar to GABA, but produces a mixture of partially-locked and desensitized states, likely underlying weaker agonist activity. Together, these results elucidate interactions of a ρ-type GABA$_A$ receptor with therapeutic drugs, offering mechanistic insights and a basis for further pharmaceutical development.

γ-Aminobutyric acid (GABA) is the major inhibitory neurotransmitter in the human central nervous system, and is crucial for the balance of neuronal excitation and inhibition[1]. Two types of GABA receptors have been identified based on their mechanism of action: GABA$_A$ receptors are GABA-gated chloride ion channels, while GABA$_B$ receptors are G protein-coupled receptors[1]. GABA$_A$ receptors belong to the pentameric ligand-gated ion channel (pLGIC) superfamily, which also contains ionotropic receptors for acetylcholine, glycine, and serotonin[2]. Channels in this family share a common architecture, with each of the five subunits contributing to an extracellular domain (ECD) with 10 strands (β1–β10) and a transmembrane domain (TMD) with 4 helices (M1–M4). The orthosteric ligand-binding site is located at the subunit interface in the ECD, composed of loops A, B and C from the principal subunit and D, E and F from the complementary subunit. In a generalized gating mechanism for pLGICs, binding of agonist to the unliganded resting state induces lockdown of loop C over the orthosteric site, and a long-range conformational transition spanning most of the protein[2]. This transition is thought to progress through a primed state

with increased ligand affinity but transient kinetics, an open state with an expanded hydrophobic gate at the midpoint across the membrane, and a desensitized state with a contracted gate at the intracellular end of the pore. Although open states of common GABA$_A$-receptor subtypes have proved difficult to resolve, a growing catalog of structures representing resting, desensitized, and intermediate states offer critical insights into conformational cycling and ligand modulation, as outlined below. In contrast, GABA$_B$ receptors mediate relatively slow and prolonged synaptic inhibition, with presynaptic GABA$_B$ receptors suppressing neurotransmitter release, and postsynaptic GABA$_B$ receptors causing hyperpolarization of neurons[3].

In humans, GABA$_A$ receptors are homo- or hetero-pentamers formed from a selection of 19 different subunits (α1-6, β1-3, γ1-3, ρ1-3, δ, ε, π and θ)[4]. Channels formed from ρ subunits were previously named GABA$_C$ receptors due to their distinctive physiological and pharmacological properties, including higher sensitivity to but slower activation by GABA relative to classical synaptic and extrasynaptic subtypes; this subtype is also relatively insensitive to bicuculline, barbiturates and

[1]Department of Applied Physics, Science for Life Laboratory, KTH Royal Institute of Technology, Solna, Sweden. [2]Department of Biochemistry and Biophysics, Science for Life Laboratory, Stockholm University, Solna, Sweden. [3]Present address: Department of Pharmacology and Chemical Biology, Shanghai Jiao Tong University School of Medicine, Shanghai, China. ✉e-mail: chf4001@kth.se; rebecca.howard@dbb.su.se; erik.lindahl@dbb.su.se

benzodiazepines[5], but sensitive to phosphinic acid compounds including (1,2,5,6-tetrahydropyridin-4-yl)methylphosphinic acid (TPMPA)[6], which provides possibilities to selectively modulate particular subforms of human GABA$_A$ receptors. Of the 3 types of ρ subunits found in mammals, ρ1 is expressed particularly in the retina, while ρ2 and ρ3 are widely distributed in brain[7,8]. ρ-type receptors play important roles in physiological processes including visual transduction[9], postnatal neurodevelopment[10], pain sensation[11], and sleep-wake cycles[12], and are potential therapeutic targets in myopia, sleep disorders, learning and memory disruption, peripheral nociception and anxiety[13,14].

Several ligands act on ρ-type GABA$_A$ receptors, including the synthetic agents 4,5,6,7-tetrahydroisoxazolo[5,4-c]pyridin-3-ol (THIP, also known as gaboxadol) and 3-aminopropyl-*n*-butylphosphinic acid (CGP36742, also known as SGS742) and the natural product γ-amino-β-hydroxybutyric acid (GABOB, also known as buxamine) (Supplementary Fig. 1). THIP, a conformationally constrained derivative of muscimol, was developed as a non-opioid analgesic and antinociceptive agent[15,16], and was a candidate in clinical trials for the treatment of insomnia[17], Fragile X syndrome[18] and Angelman syndrome[19]. CGP36742 was the first GABA$_B$ receptor antagonist in clinical trials[20], but is also an orally active antagonist of ρ-type GABA$_A$ receptors[21,22], and has shown therapeutic potential for the treatment of cognitive deficits[23,24]. As a metabolite of GABA, GABOB is found endogenously in the mammalian central nervous system, but it is also an anticonvulsant used in the treatment of epilepsy[25]. The hydroxyl group at the C3 position of GABOB generates a stereogenic center, resulting in R and S enantiomeric forms; although (R)-GABOB is a modestly more potent anticonvulsant, the compound is applied clinically as a racemic mixture[26]. Despite their therapeutic relevance, the structural foundations of these drugs' selectivity and other functional properties at ρ-type GABA$_A$ receptors remain unclear.

Here, combining four original cryo-EM structures, electrophysiology recordings, and molecular dynamics (MD) simulations, we characterize the binding and structural impact of THIP, CGP36742 and GABOB on human ρ1 GABA$_A$ receptors. We identify a distinctive binding pose of THIP in ρ1 versus the extrasynaptic neuronal α4β3δ GABA$_A$ receptors, offering a rationale for its inverse effects on these subtypes. CGP36742 binding is similar to that of TPMPA, detailing a shared mechanism of action among phosphinic acid inhibitors. In contrast, GABOB binding is similar to that of GABA, but under equivalent conditions produces a mixture of partially-locked and desensitized states; its density is compatible with both enantiomeric forms, likely representing the racemic mixture. Together, these results elucidate detailed interactions of a ρ-type GABA$_A$ receptor with therapeutic drugs, offering mechanistic insights and a prospective basis for further drug development.

## Results

### ρ1 GABA$_A$ receptors resolved with antagonist and agonist drugs

We implemented a modified human ρ1 GABA$_A$ receptor construct (ρ1-EM) with truncated loops in the N-terminus and intracellular domain and an inserted fluorescent protein to facilitate expression while largely preserving wild-type function[27]. In *Xenopus laevis* oocytes expressing ρ1-EM, THIP and CGP36742 antagonized GABA activation (Fig. 1a, b), consistent with previous studies of wild-type channels[28,29]. GABOB functioned as an agonist, albeit at ~10-fold higher concentrations than GABA (Fig. 1c), consistent with its relatively weaker activity[30].

To gain insight into the binding and conformational changes associated with these drugs, we solved cryogenic electron microscopy (cryo-EM) structures of ρ1-EM in complex with THIP, CGP36742 and GABOB in saposin nanodiscs with polar brain lipids to resolutions 2.0–2.4 Å (Fig. 1, Table 1, Supplementary Figs. 2–4). We observed non-protein densities corresponding to the expected drugs in the five extracellular orthosteric ligand-binding sites of each structure; the corresponding maps enabled us to unambiguously build models for both the protein and drugs (Fig. 1, Supplementary Fig. 5).

Consistent with their functional roles as competitive inhibitors, THIP- and CGP36742-bound structures of ρ1-EM adopted the resting-like state, nearly identical to our previously reported apo structure[27] (Table 2). In the presence of the agonist GABOB, one class (13% of resolved particles) corresponded to the desensitized state previously reported with GABA[27] (Table 2). In another class (87% of resolved particles), the ECD was not fully activated, with loop C only partially locked over the agonist site; the pore remained at rest. This structure, corresponding to neither resting nor desensitized states, aligned well with a previous so-called primed state determined in the presence of the negative modulator 17β-estradiol (E2)[31] (Supplementary Fig. 4e, f, Table 2). Notably, E2 binds in a pocket located at the ECD-TMD interface, where it appears to disrupt allosteric transitions induced by GABA binding, including lockdown of loop C over the agonist site. In contrast, we observed no ligand density at the equivalent interface in the GABOB-bound structures, indicating a distinct mechanism of stabilizing the partially-locked state.

### Distinct poses implicated in subtype-dependent THIP effects

In our cryo-EM structure, THIP was bound in the orthosteric site with its two heterocycles perpendicular to loop C (Fig. 2a). The pyridine ring faced the principal subunit, with its amino group buried in an aromatic cage involving residues Y219, Y262 and Y268. The isoxazole ring faced the complementary subunit, with its hydroxyl and amino groups making polar interactions with principal residues S264 and T265, as well as complementary residues R125 and S189 (Fig. 2a, b, Supplementary Fig. 6).

Superposing the resting-like THIP complex with our previous GABA-bound desensitized structure[27] (Fig. 2c, d) highlighted structural changes induced by antagonist versus agonist binding. The amino and hydroxyl groups of THIP roughly overlapped those of GABA; however, the bulky heterocycles of THIP were relatively protruded toward loop C, obstructing its lockdown over the antagonist. Moreover, computational docking of THIP into these two ρ1 structures produced more favorable binding energy scores in the resting-like versus desensitized state (Supplementary Fig. 7a, b). Thus, similar to TPMPA[27], THIP appears to sterically occlude local activating transitions in the ρ-type orthosteric site.

Whereas THIP is a competitive antagonist of ρ1[32] (Fig. 1a), it is a partial agonist of α1β3γ2 GABA$_A$ receptors associated with the postsynapse, and a super-agonist of α4β3δ GABA$_A$ receptors found extrasynaptically[33–35]. Consistent with this behavior, a recent cryo-EM structure of the α4β3δ subtype was reported with THIP in an apparent desensitized state, with ligands at both β3(+)-α4(-) and δ(+)-β3(-) interfaces[36]. At both these interface types, THIP adopts a pose distinct from that in the ρ1 resting state, with the heterocycles rotated nearly 90° to lie parallel to the plane of loop C (Fig. 2e, f). This THIP pose in α4β3δ GABA$_A$ receptors is compatible with more extensive lockdown of loop C than in ρ1, apparently enabling activation. We previously observed that loop C is extended by one residue in ρ1 relative to β GABA$_A$-receptor subunits, and that truncating the amino acid at the tip of loop C (ΔS264) results in functional properties more similar to postsynaptic subtypes[27]. THIP effects were similarly decreased in the ΔS264 variant, but remained inhibitory (Supplementary Fig. 8), indicating that factors other than loop C length are involved in subtype-specific modulation. Although the structural basis for the distinct binding pose is not entirely clear, several bulky groups in the ρ1 orthosteric site are substituted with smaller sidechains in β3, including Y262 (β3-F200) on the principal subunit and R125 (β3-Q64) and S189 (β3-G127) on the complementary subunit (Supplementary Fig. 6).

### Structural basis for specificity among phosphinic acids

In our cryo-EM structure, CGP36742 adopted an elongated pose, with its aminopropyl tail forming a salt bridge with E217 and cation-π interactions with the aromatic cage on the principal subunit face. The

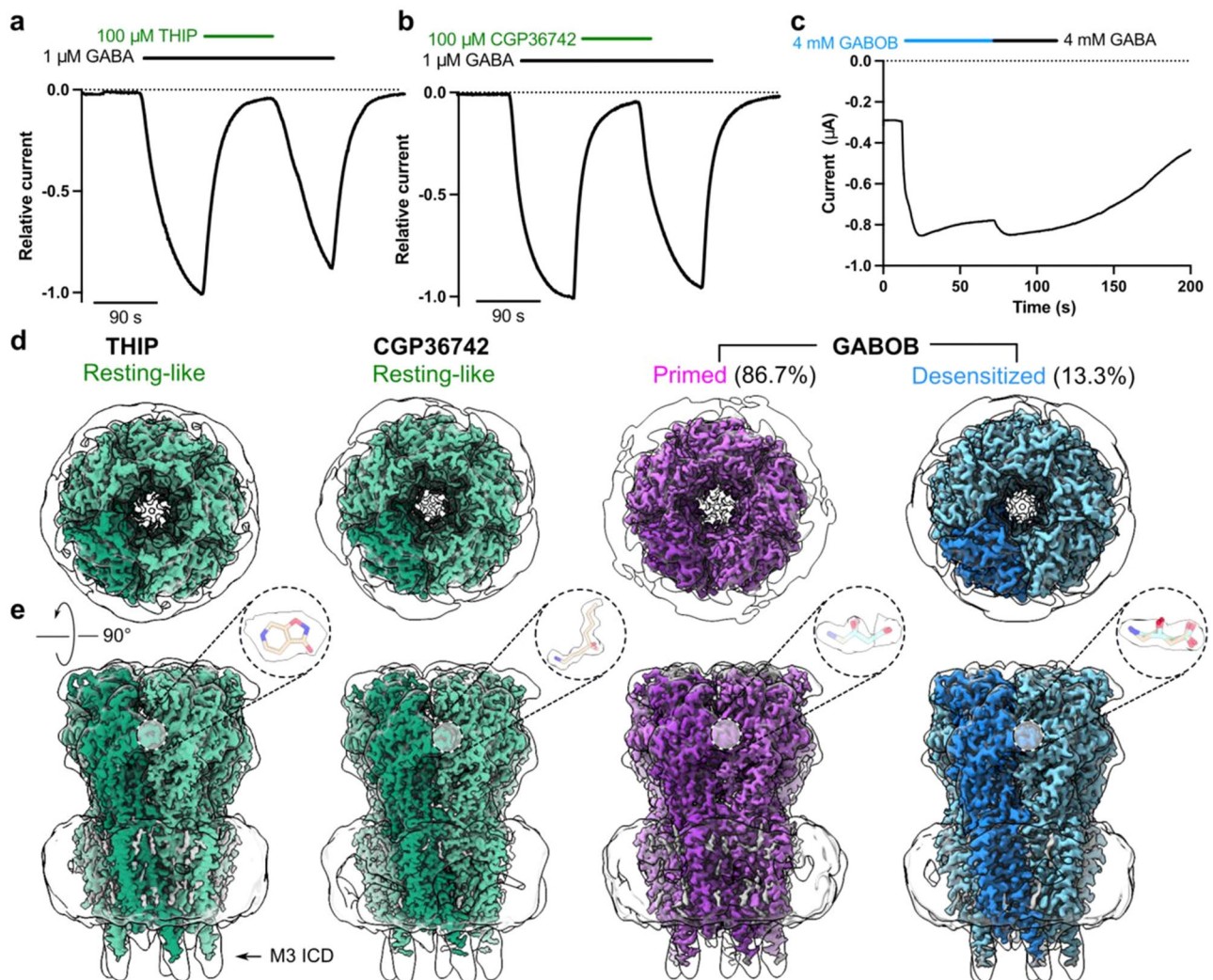

**Fig. 1 | Functional and structural profiles of ρ1 with antagonist and agonist drugs. a–c** Sample traces from two-electrode voltage-clamp electrophysiology recordings of ρ1-EM expressed in *Xenopus oocytes* exposed to THIP, CGP36742 or GABOB. **d** Cryo-EM maps of human ρ1-EM in complex with THIP (left), CGP36742 (middle) or GABOB (right), viewed from the extracellular side. Maps are colored by assigned functional states as similar to resting (green), partially locked (purple), or desensitized (blue). Low-pass filtered maps are shown in transparency to reveal lower-resolution features including the nanodisc and M3-helix intracellular-domain extension (M3 ICD). **e** Cryo-EM maps as in (**d**), viewed from the membrane plane. Insets show the drugs with corresponding densities.

central phosphinic acid group was wedged between loop C on the principal subunit (making polar interactions with S264 and T265) and residue R125 on the complementary subunit, with the butyl tail extending toward the complementary β5 strand (Fig. 3a, b). The complex was comparable to the apo and THIP structures (Table 2, Fig. 3c), and superimposable with our previous structure with TPMPA (Cα RMSD 0.035 Å) (Fig. 3d), including analogous interactions of the amino and phosphinic acid groups; the additional butyl tail only required an alternative rotamer of M177 in the β5 strand (Fig. 3c, d). As in the case of TPMPA, the bulky, electronegative phosphinic acid group appeared to prevent lockdown of loop C over the antagonist, indicating a shared mechanism as well as binding pose among phosphinic acid inhibitors (Fig. 3e, f).

Given the ability of CGP36742 to inhibit GABA$_B$ as well as ρ1 GABA$_A$ receptors[29,37], we also compared its potential interactions with these structurally distinct targets. A structure of the related phosphinic acid inhibitor CGP35348 was previously reported with the ECD of a human GABA$_B$ receptor, bound in the cleft between ligand-binding lobes (LB1 and LB2) of the GBR1 subunit[38] (Fig. 4a–c). The CGP36742 pose from our ρ1-EM structure could be placed in this pocket in the GABA$_B$ receptor complex by superposition on its equivalent aminopropyl and phosphinic acid moieties (Fig. 4b), or by computational docking (Supplementary Fig. 7c), with similar poses showing no evident clashes and a favorable binding energy score. Interestingly, the amino group of CGP36742 was oriented roughly opposite to that of CGP35348 in its resolved site (Fig. 4d), indicating the rotational flexibility of these compounds could support their polymodal activities.

The most potent and selective ρ1 antagonist identified thus far, (4-aminocyclopenten-1-yl)-butylphosphinic acid ((S)−4-ACPBPA), shares the phosphinic acid and butyl groups of CGP36742 but has a conformationally restricted aminocyclopentenyl group in place of the flexible aminopropyl tail[39,40] (Supplementary Fig. 1). Superposition of (S)−4-ACPBPA with CGP36742 in our ρ1-EM complex showed this ligand could be accommodated without modification (Fig. 4e). Conversely, superposing this compound with CGP35348 in the GABA$_B$ receptor resulted in a clash of the amino group with residue W65, suggesting a molecular basis for receptor specificity (Fig. 4f). Consistent with these predictions, computational docking of (S)−4ACPBPA produced more favorable binding energy scores in ρ1-EM than in the GABA$_B$ receptor (Supplementary Fig. 7d, e).

**Multiple states captured with the weak racemic agonist GABOB**

As described above, our cryo-EM dataset collected with GABOB contained particles in both partially-locked and desensitized states.

**Table 1 | Cryo-EM data collection, refinement and validation statistics**

| PDB ID<br>Ligand<br>Apparent state | 9FRE<br>THIP<br>Resting-like | 9FRB<br>CGP<br>Resting-like | 9FRF<br>(R)-GABOB<br>Partially locked | 9FRI<br>(S)-GABOB<br>Partially locked | 9FRF<br>(R)-GABOB<br>De-sensitized | 9FRG<br>(S)-GABOB<br>De-sensitized |
|---|---|---|---|---|---|---|
| Data collection and processing | | | | | | |
| Magnification | 130,000 | 130,000 | 130,000 | | | |
| Voltage (kV) | 300 | 300 | 300 | | | |
| Electron exposure ($e^-/Å^2$) | 48.3 | 44.0 | 47.0 | | | |
| Defocus range (µm) | −0.8 to −1.8 | −0.8 to −1.8 | −0.8 to −1.8 | | | |
| Pixel size (Å) | 0.6725 | 0.6725 | 0.67 | | | |
| Symmetry imposed | C5 | C5 | C5 | | C5 | |
| Final particles | 135,764 | 398,556 | 191,614 | | 57,366 | |
| Map resolution (Å) | 2.19 | 2.05 | 2.14 | | 2.41 | |
| FSC threshold | 0.143 | 0.143 | 0.143 | | 0.143 | |
| Refinement | | | | | | |
| Map sharpening $B$ factor ($Å^2$) | −66.2 | −63.6 | −66 | | −70.8 | |
| Non-hydrogen atoms | 14,491 | 14,402 | 13,756 | 13,756 | 13,882 | 13,882 |
| Protein residues | 1670 | 1665 | 1660 | 1660 | 1645 | 1645 |
| Ligands | 56 | 57 | 56 | 56 | 42 | 42 |
| $B$ factors ($Å^2$) | | | | | | |
| Protein | 31.59 | 10.02 | 14.49 | 11.03 | 41.01 | 41.01 |
| Ligand | 57.41 | 24.84 | 32.35 | 24.08 | 71.63 | 70.76 |
| R.m.s. deviations | | | | | | |
| Bond lengths (Å) | 0.004 | 0.007 | 0.006 | 0.006 | 0.007 | 0.007 |
| Bond angles (°) | 0.879 | 0.828 | 1.211 | 1.219 | 1.168 | 1.171 |
| Validation | | | | | | |
| MolProbity score | 1.32 | 0.99 | 1.68 | 1.68 | 0.88 | 0.88 |
| Clashscore | 4.51 | 2.14 | 5.41 | 5.41 | 0.58 | 0.58 |
| Poor rotamers (%) | 0.00 | 0.00 | 0.63 | 0.7 | 0.00 | 0.00 |
| Ramachandran plot | | | | | | |
| Favored (%) | 97.58 | 98.48 | 94.21 | 94.21 | 96.92 | 96.92 |
| Allowed (%) | 2.42 | 1.52 | 5.49 | 5.49 | 3.08 | 3.08 |
| Disallowed (%) | 0.00 | 0.00 | 0.30 | 0.30 | 0.00 | 0.00 |

**Table 2 | Cα RMSD (Å) between ρ1-EM structures**

| | THIP<br>Resting-like | CGP36742<br>Resting-like | GABOB<br>Partially locked | GABOB<br>Desensitized |
|---|---|---|---|---|
| Apo resting (PDB 8OQ6) | 0.28 | 0.34 | 0.64 | 1.69 |
| GABA + E2 partially locked (PDB 8RH7) | 0.7 | 0.71 | 0.27 | 1.38 |
| GABA desensitized (PDB 8RH8) | 1.75 | 1.78 | 1.44 | 0.06 |

The partially-locked state exhibited only a partial lockdown of loop C over the ligand, corresponding to a limited (1.8 Å) S264-Cα translation, and a subtle (1.2°) domain rotation relative to the resting state (Fig. 5a). Although the contribution of this partially-locked state to the receptor gating cycle remains unclear, the TMD was superimposable with that of the resting state, consistent with it representing a pre-active intermediate between resting and open. The presence of a substantial partially-locked class with GABOB may reflect the relatively low affinity and slow kinetics of this agonist, despite the application of a supersaturating concentration (4 mM) for >30 min prior to grid freezing.

Consistent with our previous structure with GABA[27], the desensitized state with GABOB exhibited further lockdown of loop C (4.3 Å S264-Cα translation) and rotation between the ECD and TMD (7.3°) relative to the partially-locked state (Fig. 5b). Interestingly, although all structures in this work contained ligands in the extracellular orthosteric site, local resolution in the ECD was relatively higher in the GABOB-desensitized state; in resting and partially-locked structures, resolution was similar between the domains (Supplementary Fig. 4a–d), suggesting that channel activation is associated with both stabilization of the ECD and mobilization of the TMD.

Densities for GABOB were clearly resolved in both the partially-locked and desensitized states (Fig. 6a, b). Although the C3 carbon of GABOB constitutes a stereocenter, both enantiomers are ρ1 agonists, with modestly greater potency for (R)-GABOB[30]. As in medical practice, we used a racemic mixture in our experiments, such that our cryo-EM densities likely contained both forms; indeed, either (R)- or (S)-GABOB could be modeled into the ligand density in either structure (Fig. 6a, b). The amino end was coordinated by an aromatic cage (Y219, Y262, Y268) as well as by E217 in the principal subunit, while the carboxylate end made electrostatic contacts with R125 and S189 in the complementary subunit (Fig. 6c–f). In all cases, T265 at the tip of loop C could make a hydrogen bond with the C3 hydroxyl, consistent with the demonstrated effect of this residue on potency of both GABOB forms[41].

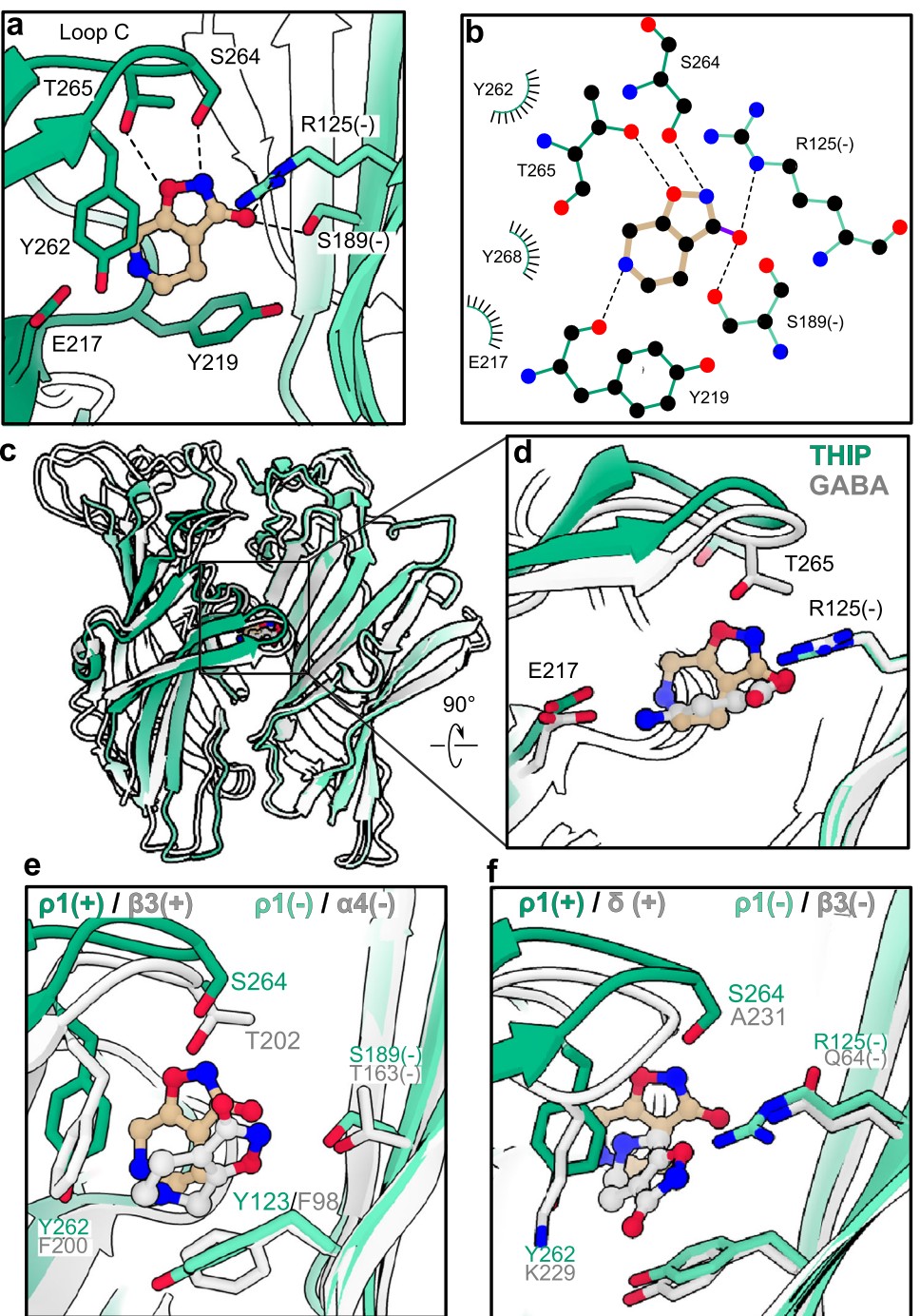

**Fig. 2 | Distinctive binding pose of THIP in the ρ1 GABA_A receptor. a** Zoom view of the THIP (tan) binding site in ρ1-EM (green). Residues interacting directly with THIP are shown as sticks and colored by heteroatom; residues from the complementary subunit face are denoted (-). Potential hydrogen bonds are indicated as dashed lines. **b** Schematic of ρ1-EM interactions with THIP, colored as in (**a**). Hydrogen bonds are indicated as dashed lines, hydrophobic and aromatic interactions as lashes. **c, d** Superimposition of the ECD of THIP-bound (green) and GABA-bound (PDB ID 8OP9, gray) ρ1-EM structures. For clarity, only two subunits are shown, aligned on the complementary subunit. Ligands are shown as sticks, with carbon atoms of THIP and GABA colored tan and gray respectively. **e, f** Zoom views as in (**d**) of the superimposition of the THIP binding site in ρ1-EM (green) and the α4β3δ GABA_A receptor (PDB ID 7QND, gray), focusing on the β3-α4 or δ-β3 interfaces, respectively. Structures are aligned on the ECD of the complementary subunit. Ligands are shown as sticks, with THIP carbon atoms in ρ1 and α4β3δ subtypes colored tan and gray respectively.

To substantiate binding capacity for both entantiomers, we performed all-atom molecular dynamics (MD) simulations of both the partially-locked and desensitized-state structures with both (R)- and (S)-GABOB. In ≥400-ns simulations of each system, performed in quadruplicate with different initial velocities, both (R)- and (S)-GABOB remained within 2 Å median root-mean-square deviation (RMSD) relative to their starting poses in both structures (Fig. 6g,

Supplementary Fig. 9), consistent with weak enantiomeric specificity on the timescale of atomistic simulations. Previously we have also observed an intersubunit hydrogen bond between residue Y268 on the principal loop C and R179(-) on the complementary loop F, which characterizes ligand-bound versus -unbound states[27] (Fig. 6h). Similar to our previous simulations with GABA, either of the GABOB enantiomers sustained tighter Y268-R179 interactions than in the resting

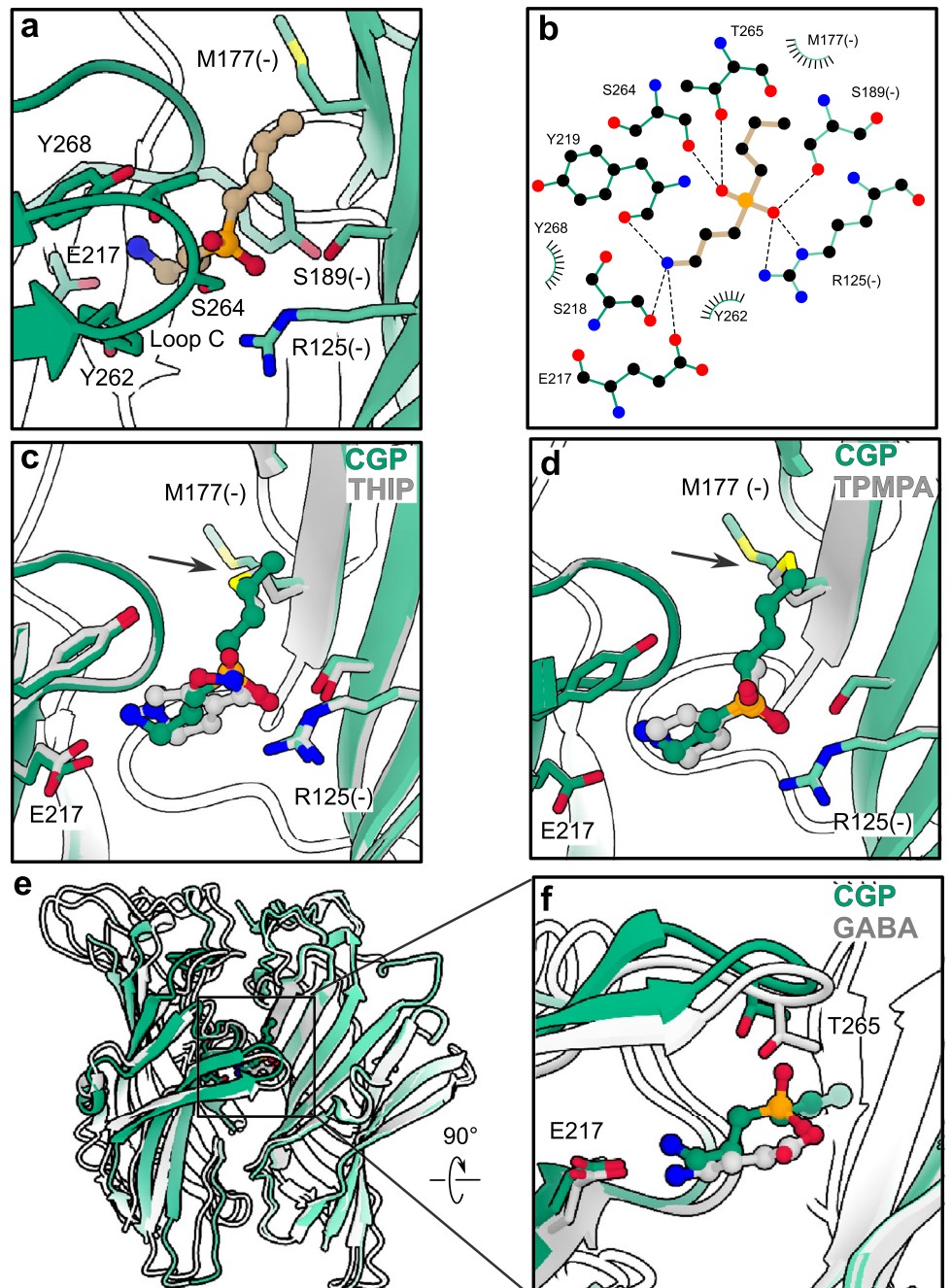

**Fig. 3 | Common mechanism of ρ1 GABA$_A$ receptor antagonism by phosphinic acid inhibitors. a** Zoom view of CGP36742 (tan) binding site in ρ1-EM (green), colored and labeled as in Fig.2a. **b** Schematic of ρ1-EM interactions with CGP36742. Hydrogen bonds and other electrostatic interactions are indicated as dashed lines, hydrophobic and aromatic interactions as lashes. **c** Zoom view of superimposed structures of ρ1-EM with CGP36742 (green) and THIP (gray). **d** Zoom view of superimposed structures of ρ1-EM with CGP36742 (green) and TPMPA (PDB ID 8OQ7, gray). **e, f** Superimposition of the ECD of CGP36742-bound (green) and GABA-bound (PDB ID 8OP9, gray) ρ1-EM structures. For clarity, only two subunits are shown, aligned on the complementary subunit.

structure, though a modest trend towards even tighter contacts in the presence of (R)-GABOB appeared consistent with the slightly greater potency of this enantiomer (Fig. 6i). Interestingly, this interaction was retained in the partially-locked as well as desensitized systems, despite the limited extent of ECD activation.

## Discussion

Pharmaceutical targeting of ρ-type GABA$_A$ receptors holds promise for drug development in treating visual, sleep, learning and memory disorders[13,42]. Given the limited sensitivity of this subtype to classical GABA$_A$ receptor modulators such as benzodiazepines, barbiturates and general anesthetics, the development of such agents will likely require a detailed understanding of ρ-specific mechanisms including binding, activation and inhibition. However, the specific mechanisms also means such drugs could limit interactions on neuronal receptors. The structural, functional and computational work presented here uncovers the binding modes of three drugs in the orthosteric site, highlighting among other things the critical role of loop C lockdown in channel gating.

Our structures highlight the critical role for loop-C lockdown in initiating ρ1 activation. Antagonists such as THIP and phosphinic acids clearly obstruct lockdown of loop C altogether, resulting in an

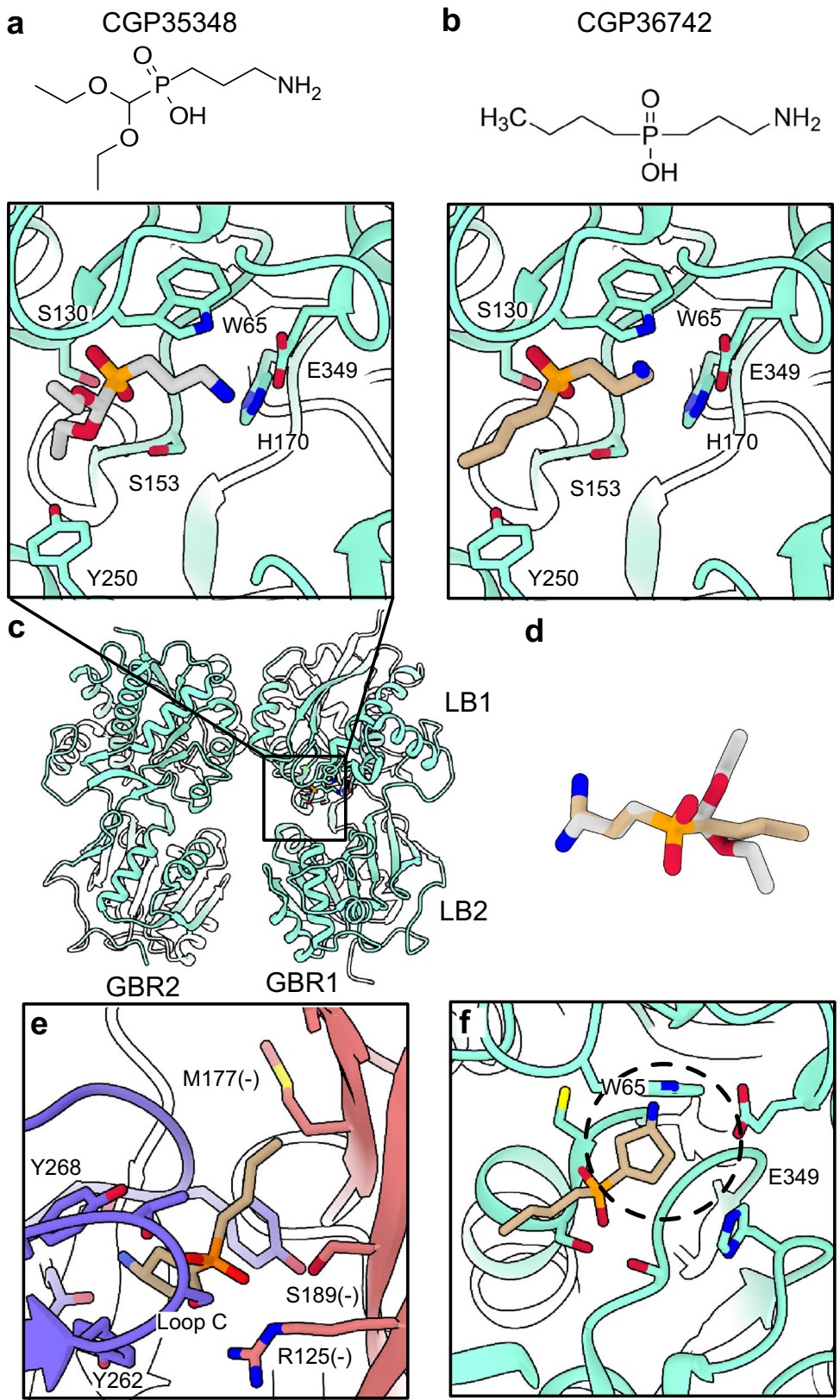

**Fig. 4 | Comparative interactions of GABA_B and GABA_A receptors with phosphinic acid inhibitors. a** Chemical structure of CGP35348 (above) and its binding site in a human GABA_B receptor (PDB ID 4MR8, green, below). The ligand (gray) and its direct residue contacts are colored by heteroatom. **b** Chemical structure of CGP36742 (above), and its superimposition into the equivalent GABA_B-receptor site as in (**a**). The ligand pose is adopted from the structure reported here with ρ1-EM, aligned on the shared aminopropyl and phosphinic acid moieties of CGP35348. **c** Overview of the ECD of the human GABA_B receptor as shown in (**a**). **d** Alignment of CGP36742 in ρ1-EM (tan) with CGP35348 in the GABA_B receptor (gray) as implemented in *b*, colored by heteroatom. **e** Alignment of (S)−4-ACPBPA (tan) into the CGP36742 site of ρ1-EM. For perspective, principal and complementary subunits of ρ1-EM are colored purple and pink respectively; the ligand and interacting residues are colored by heteroatom. **f** Alignment of (S)−4-ACPBPA (tan) into the CGP35348 of the GABA_B receptor site (green) shown in *a*. Dashed circle indicates prospective clash with residue W65.

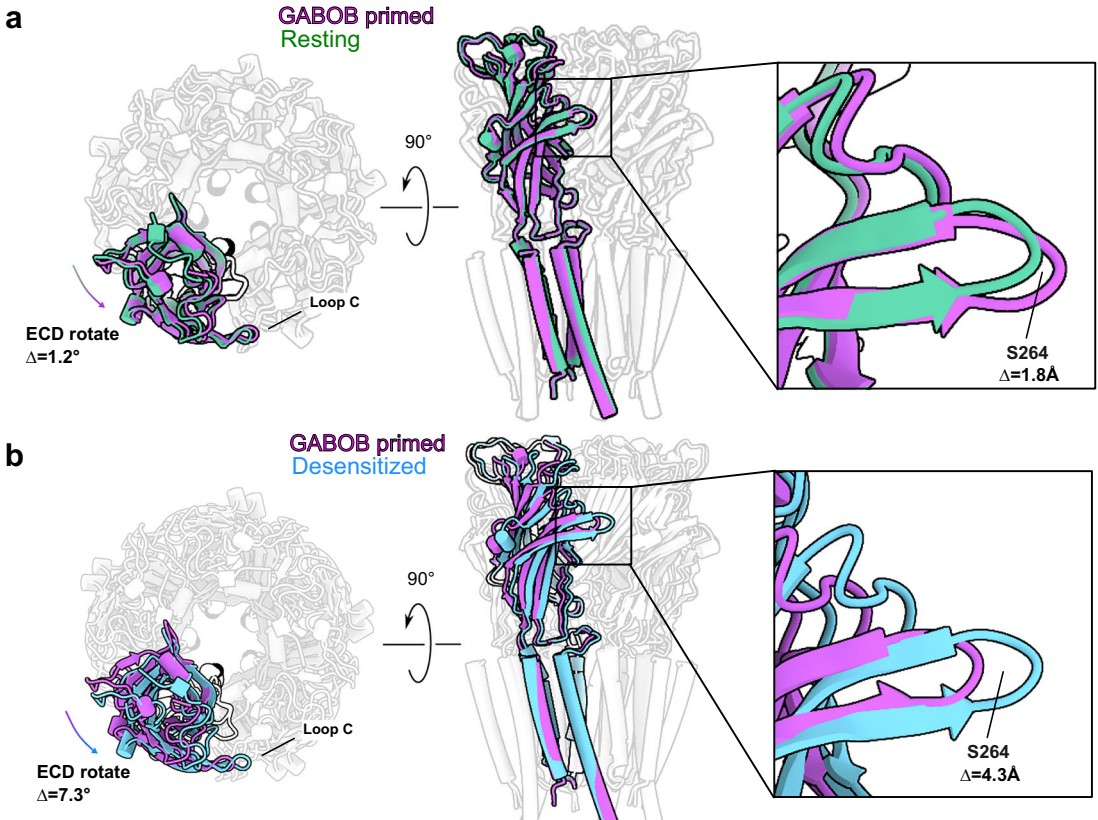

**Fig. 5 | Partially-locked and desensitized states captured with GABOB.**
**a** Superimposed structures of the GABOB-bound partially-locked state (purple) with a previously reported resting state (green, PDB 8OQ6). Structures are aligned on the TMD; for clarity, all but one of the subunits are semi-transparent. Labeled distance in the zoomed figure is between S264 Cα positions in the two structures.

The S264 side chain is shown as sticks, including polar hydrogens, with oxygen and hydrogen colored red and gray respectively. **b** Superimposed structures of GABOB-bound partially-locked state (purple) and desensitization state (blue). The structures are aligned with the TMD domain, and for clarity all but one of the subunits are transparent.

expansive orthosteric site superimposable with that of the apo structure (Fig. 7). Among phosphinic acid inhibitors, lockdown is obstructed by a consistent binding pose of CGP36742 and TPMPA, and likely of the subtype-specific agent (S)−4-ACPBPA. Conversely, cross-reactivity of CGP36742 with GABAB receptors is attributable at least in part to rotational flexibility around the amino group. Despite the shared binding profile among phosphinic acid inhibitors in both GABA-receptor families, modification of the aminopropyl tail in (S)−4-ACPBPA, or of the butyl tail in CGP35348, successfully confers preference for GABA$_A$ and GABA$_B$ receptors respectively, indicating that selective modulators can be engineered on this scaffold.

At the other extreme, agonists such as GABA enable substantial lockdown of loop C, resulting in compaction of the orthosteric site and rotation of the ECD relative to the TMD (Fig. 7a, b). Alongside this apparent activated-desensitized state, treatment with the weaker agonist GABOB promoted a subclass exhibiting only partial lockdown of loop C relative to the apo form (Fig. 7a, c). We previously reported a similar partially-locked structure in the presence of the inhibitor E2, which might allosterically trap the so-called primed state, or some metastable intermediate on the pathway from resting to open[31]. In the absence of allosteric inhibition, GABOB appears to favor a partially-locked population by different means. Given its low potency and slow kinetics relative to GABA, GABOB occupancy may be insufficient to stimulate complete rotation/activation of the ECD in a subset of ρ1 particles. It is also plausible that non-physiological sample conditions, such as embedding in a lipid nanodisc or non-instantaneous plunge-freezing, may relatively destabilize the GABOB-activated state. Although agonist binding to at least 3 of 5 subunit interfaces is thought to enable

ρ1 activation[43], the limited diffusive volume and high local receptor density on the cryo-EM grid may limit ligand accessibility, even in the presence of a supersaturating bulk agonist concentration. Indeed, it remains unclear why a fully activated-open structure of ρ1 remains experimentally inaccessible with either GABA or GABOB[27,31].

Subtype- and agent-specific interactions, e.g. with T265 on loop C, suggest avenues for future structure-based drug design. For instance, we observed a distinctive binding pose for THIP in ρ1-EM compared to a previously reported complex with an α4β3δ GABA$_A$ receptor[36], which may underlie its opposing effects in these subtypes. Interestingly, the THIP derivative aza-THIP is a more selective ρ-type antagonist, with activity comparable to THIP at ρ1 but negligible at heteromeric GABA$_A$ receptors[28]. The two molecules are identical except at one heavy atom in the 5-membered ring, substituting a pyrazole in aza-THIP for the isoxazolo group in THIP (Supplementary Fig. 1a). In structures with THIP, the substituted oxygen atom appears to accept a hydrogen bond from principal-subunit loop C (T265) in ρ1-EM, but from the complementary subunit (α4-T163 or β3-Q64) in the α4β3δ GABA$_A$ receptor. The differential environments for this substituted atom in particular may account for discriminating activity between these two receptors. A hydrogen bond with loop-C T265 also appears to underlie potency of both enantiomeric forms of GABOB in ρ1, and may contribute to the stabilization of a partially-locked state in the presence of this weaker agonist relative to GABA. Taken together, this work details receptor-specific binding interactions of both antagonists and agonists in the orthosteric site, offering potential insights into differential pharmacology across multiple receptor subtypes in the GABAergic system.

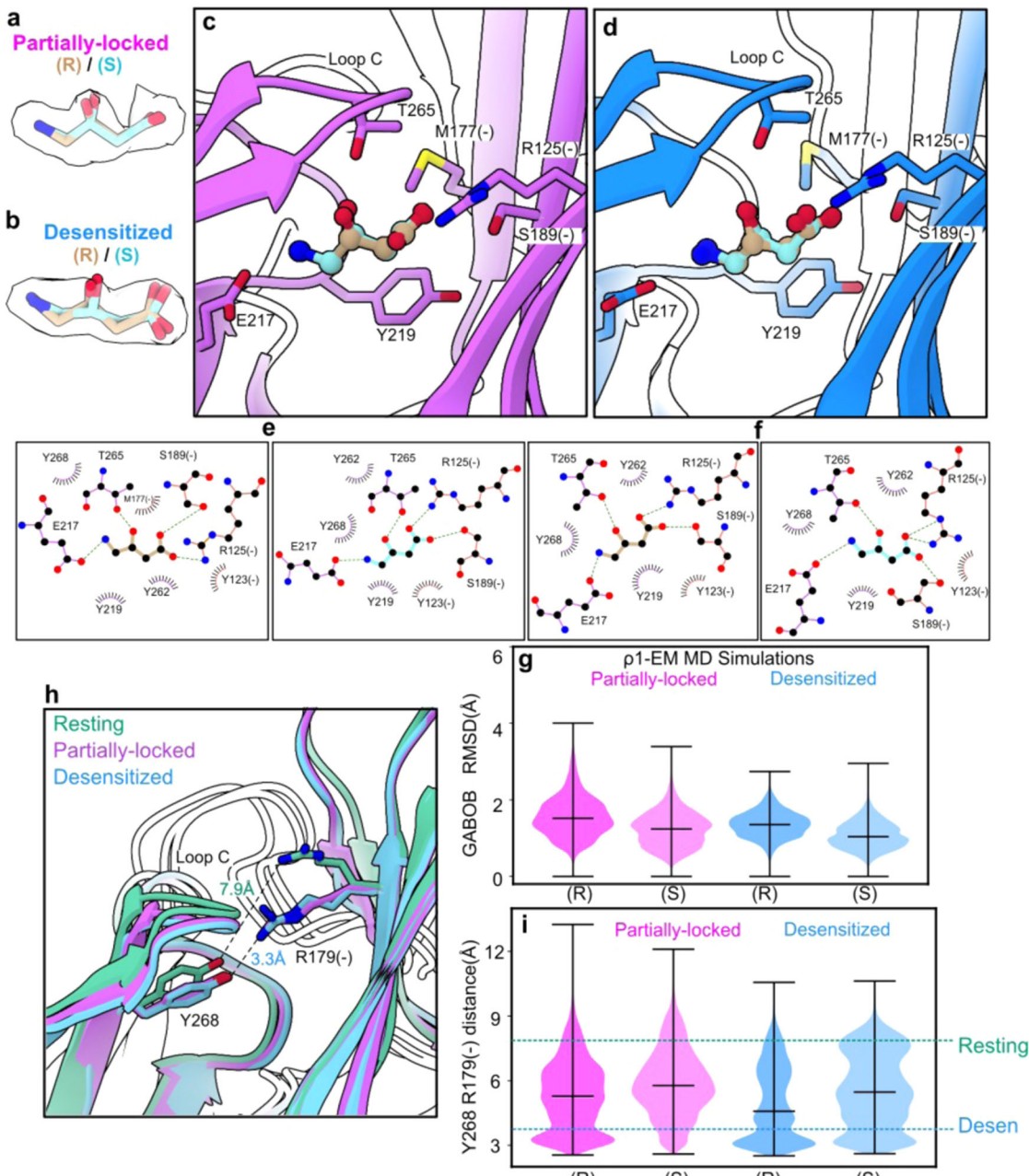

**Fig. 6 | Comparable accommodation of GABOB enantiomers in the orthosteric site. a** Refinement of (R)- (tan) and (S)-GABOB (cyan) into ligand density in the partially-locked state of ρ1-EM (purple), colored by heteroatom. **b** Refinement of (R)- and (S)-GABOB into ligand density in the desensitized state of ρ1-EM (blue), otherwise colored as in (**a**). **c** Protein-ligand interactions of (R)- and (S)-GABOB in the partially-locked state of ρ1-EM, colored as in (**a**). **d** Protein-ligand interactions of (R)- and (S)-GABOB in the desensitized state of ρ1-EM, colored as in (**b**). **e** Schematics of (R)- (left, tan) and (S)-GABOB (right, cyan) interactions with the partially-locked state of ρ1-EM. **f** Schematics of (R)- (left, tan) and (S)-GABOB (right, cyan) interactions with the desensitized state of ρ1-EM. In (**e**, **f**), hydrogen bonds and other electrostatic interactions are indicated as dashed lines, hydrophobic and aromatic interactions as lashes. **g** Mobility of (R)- and (S)-GABOB in MD simulations of ρ1-EM in the partially-locked (left, purple) and desensitized states (right, blue), as quantified by ligand RMSD (Å). Violin plots represent probability densities from 4 independent simulation replicates, sampled every 0.4 ns for the first 400 ns of each replicate (*n* = 4000), with markers indicating median and extrema. **h** Zoomed view of the orthosteric ligand site in resting (green), partially-locked (purple) and desensitized states (blue). Labels indicate the distance between nearest heavy atoms of Y268 and R179(-), where an apparent hydrogen bond characterizes ligand-bound versus -unbound states. **i** Distance between the Y268 hydroxyl and R179(-) guanidinium groups in MD simulations of ρ1-EM with (R)- and (S)-GABOB in the partially-locked (left, purple) and desensitized states (right, blue). Violin plots represent probability densities from 4 independent simulation replicates, sampled every 0.4 ns for the first 400 ns of each replicate (*n* = 4000), with markers indicating median and extrema. Dashed lines indicate static distances in the resting (green) and desensitized (blue) structures.

## Methods

### Protein purification from mammalian cells

The expression and purification of ρ1-EM was following our earlier published methods[27]. Briefly, cell pellets from 2 L of Expi293F cells infected with baculovirus were resuspended in the buffer (40 mM HEPES pH 7.5, 300 mM NaCl, with cOmplete protease inhibitor tablets (Roche)) and sonicated to break cell membranes. The membrane was pelleted by ultracentrifugation then resuspended and solubilized by resuspension buffer with 2% lauryl maltose neopentyl glycol (LMNG), 0.2% cholesteryl hemisuccinate (CHS) for 3 h in the cold room. The

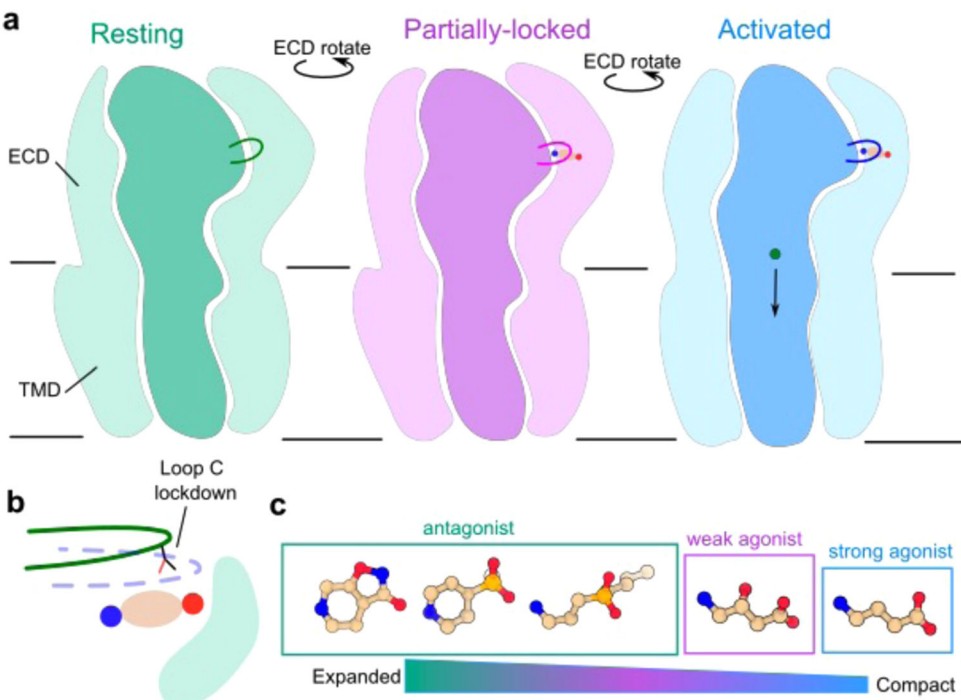

**Fig. 7 | Proposed mechanisms of antagonist and agonist drugs. a** Cartoons of ρ1 showing progressive ECD rotation between resting (green), partially-locked (purple) and activated (open or desensitized, blue) states. Loop C is highlighted, and GABA and ions are shown in circles (carbon, tan; amine, blue; carboxylate, red; chloride, green). **b** Cartoon zoom view of the orthosteric binding site. GABA is shown in circles as in (**a**). Green and blue lines depict lockdown of loop C. **c** Proposed dependence of orthosteric-site expansion/compaction on ligand identity, from antagonists to strong agonists.

solubilization mixture was ultracentrifuged and the supernatant was applied to 4 mL Strep-Tactin XT Superflow resin (IBA) and incubated for 90 min. Resin was washed with wash buffer (20 mM HEPES pH 7.5, 300 mM NaCl, 0.005% LMNG, 0.0005% CHS) then protein was eluted with elution buffer (wash buffer with 10 mM d-Desthiobiotin (Sigma)). The product was further purified by size exclusion chromatography on a Superose 6 column (Cytiva) with flow buffer (20 mM HEPES pH 7.5, 100 mM NaCl, 0.005% LMNG, 0.0005% CHS). Peak fractions were pooled for nanodisc reconstitution.

### Nanodisc reconstitution
The plasmid for SapA expression was a gift from Salipro Biotech AB. Purification of SapA followed the published protocols[44]. For the reconstitution of nanodisc, ρ1-EM, SapA and polar brain lipid (Avanti) were mixed as molar ratio 1:15:150, then incubated on ice for 1 h. Bio-Beads SM-2 resin (Bio-Rad) was added into the mixture then gently rotated overnight at 4 °C. On the next day, the supernatant was collected and further purified by gel-filtration chromatography on a Superose 6 column (Cytiva) with buffer containing 20 mM HEPES pH 7.5, 100 mM NaCl. Peak fractions were pooled and concentrated to ~5 mg/mL.

### Cryo-EM grid preparation and data collection
The nanodisc sample was mixed with the compound stock solutions with volume ratio 9:1. The stock solutions were (5 mM THIP, 20 mM fluorinated foscholine 8 (FFC-8)), (20 mM CGP36742, 20 mM FFC-8), (40 mM GABOB, 20 mM FFC-8). The mixtures were incubated on ice for more than 30 minutes before freezing grids. For each grid, 3 µL of the mixture was applied to a glow-discharged grid (R1.2/1.3 300 mesh Au grid, Quantifoil), blotted for 2 s with force 0 and plunged into liquid ethane using a Vitrobot Mark IV (Thermo Fisher Scientific). Cryo-EM data were collected on a 300 kV Titan Krios (Thermo Fisher Scientific) electron microscope with a K3 Summit detector (Gatan) with magnification 130k corresponding to 0.6725 or 0.6645 Å/px using the

software EPU 3.5.0 (Thermo Fisher Scientific). The total dose was ~46 e⁻/Å² and defocus range was −0.8 to −1.8 µm.

### Cryo-EM data processing
Dose-fractionated images in super-resolution mode were internally gain-normalized and binned by 2 in EPU during data collection. Cryo-EM data processing was first done in Relion 3.1.4[45], including Motion correction, contrast transfer functions (CTF) estimation with CTFFIND 4.1[46], automatic particle picking with topaz 0.2.5[47], particle extraction, 2D classification, 3D classification, 3D refinement, CTF refinement and polishing. Briefly, two rounds of 2D classification were done to remove junk particles, 3D classification (3 classes) was used to analyze the structural heterogeneity. Particles from classes with protein features were centered and re-extracted, and were used for the 3D refinement with symmetry C5. Several rounds of CtfRefine and one round of polishing were executed to improve the resolution. The shiny particles were imported into CryoSPARC v4.2.1 for further processing[48], including 3D classification with the PCA mode, and Non-Uniform Refinement[49].

### Model building and refinement
Model building was started with rigid body fitting of the previously published resting (PDB ID 8OQ6), primed (PDB ID 8RH7) or desensitized (PDB ID 8RH8) state structure into the density. The models were manually checked and adjusted in Coot 0.9.5[50], and chemicals, waters and lipids were also manually added. The resulting model was further optimized using real-space refinement in PHENIX 1.18.2[51] and validated by MolProbity[52]. Crystallographic information files (cif) for ligands were generated from isomeric SMILES strings using Grade2[53]. For initial ligand comparisons across receptor families, ligands (CGP36742, (S)−4-ACPBPA) were superposed using the "align" command in UCSF Chimera[54] to match corresponding non-hydrogen atoms in the target complex (CGP35348, CGP36742).

## Structural analysis

Pore radius profiles were calculated using CHAP 0.9.1[55]. Structure figures were prepared using UCSF ChimeraX 1.3[56]. ECD rotation was calculated as the dihedral angle between a) the Cα COM of the ECD (residues 97-280) of one subunit, b) the equivalent ECD residues of all subunits, c) the TMD (residues 281-479) including all subunits, and d) the equivalent TMD residues of one subunit. RMSDs were calculated by aligning Cα atoms in two given structures using the "match" command in UCSF Chimera[54]. Docking was performed using Autodock Vina[57], with search volumes 15 Å * 15 Å * 15 Å around each pocket.

## Expression in oocytes and electrophysiology

mRNA encoding the ρ1-EM GABA_A receptor was produced by in-vitro transcription using the mMessage mMachine T7 Ultra transcription kit (Ambion) according to the manufacturer protocol. *Xenopus laevis* oocytes (Ecocyte Bioscience) were injected with 30–50 ng mRNA and incubated 4–8 days at 13 °C in post-injection solution (10 mM HEPES pH 8.5, 88 mM NaCl, 2.4 mM NaHCO3, 1 mM KCl, 0.91 mM CaCl2, 0.82 mM MgSO4, 0.33 mM Ca(NO3)2, 2 mM sodium pyruvate, 0.5 mM theophylline, 0.1 mM gentamicin, 17 mM streptomycin, 10,000 u/L penicillin) prior to two-electrode voltage clamp measurements.

For recordings, glass electrodes were pulled and filled with 3 M KCl to give a resistance of 0.5–1.5 MΩ and used to clamp the membrane potential of injected oocytes at −60 mV with an OC-725C voltage clamp (Warner Instruments). Oocytes were maintained under continuous perfusion with Ringer's solution (123 mM NaCl, 10 mM HEPES, 2 mM KCl, 2 mM MgSO4, 2 mM CaCl2, pH 7.5) at a flow rate around 1.5 mL/min. Buffer exchange was accomplished by manually switching the inlet of the perfusion system to the appropriate buffer. For assessing GABOB efficacy, a gravity-fed perfusion system was used to improve kinetics of solution exchange. Currents were digitized at a sampling rate of 2 kHz and lowpass filtered at 10 Hz with an Axon CNS 1440 A Digidata system controlled by pCLAMP 10 (Molecular Devices).

## Molecular dynamics simulations

All-atom simulations in explicit solvent were deemed most appropriate to assess steady-state dynamics, given the relatively high precision and accuracy of atomistic interactions that can be captured compared to e.g. coarse-grained methods. Atomic coordinates of the ρ1-GABOB complex with GABOB built as either of two enantiomers were used as starting models for MD simulations. Each subunit was split into two chains for simulation, due to unresolved residues in the M3-M4 loop in the experimental structure. This approach carries the risk of enabling non-physiological dynamics around deletion endpoints; on the other hand, it avoids introducing information without direct experimental evidence, for example by modeling the missing loop, inserting a shorter sequence, or applying positional restraints. Given that the sub-microsecond timescales of our simulations are not expected to sample allosteric effects of the intracellular domain on drug-binding sites, which are located at least 70 Å away on the opposite side of the membrane, and that overall Cα RMSDs remained within 4 Å (Supplementary Fig. 9), this approach appeared to be a reasonable and parsimonious compromise. The simulation systems were set up in CHARMM-GUI[58]. The protein was embedded into a bilayer mimicking brain-lipid composition with the top leaflet containing 155 POPC, 24 POPE, and 38 cholesterol molecules and the bottom leaflet containing 65 POPC, 115 POPE, 26 POPS, and 32 cholesterol molecules. The protein-lipid complex was subsequently solvated with TIP3P water and 150 mM NaCl (Supplementary Table 1). The CHARMM36m forcefield[59] was used to describe the protein. Parameters for (R)- and (S)-GABOB were generated using CGenFF[60] in CHARMM-GUI. Visual and quantitative inspection of the resulting parameters (param penalty 4.600, charge penalty 10.008) indicated reasonable description of both ligands. Cation-π specific NBFIX parameters were used to maintain appropriate ligand-protein interactions in the aromatic cage in the orthosteric binding site[61].

Simulations were performed using GROMACS 2022.5[62] at 300 K and 1 bar using the velocity-rescaling thermostat[63] and Parrinello–Rahman barostat[64]. The LINCS algorithm was used to constrain the length of all bonds involving hydrogens[65], and the particle mesh Ewald method[66] was used to calculate long-range electrostatic interactions. The systems were energy minimized and then equilibrated for 20 ns, with the position restraints on the protein and neurosteroids gradually released. Four replicates, each >400 ns, were simulated for each system as final unrestrained production runs. Before analysis, MD simulation trajectories were aligned on the Cα atoms of the ECD using MDAnalysis[67]. Root-mean-square deviations (RMSD) of the non-hydrogen atoms of GABOB were calculated in VMD[68], and data from the first 400 ns were combined and plotted as violin plots using Matplotlib[69].

## Reporting summary

Further information on research design is available in the Nature Portfolio Reporting Summary linked to this article.

## Data availability

The cryo-EM maps have been deposited in the Electron Microscopy Data Bank (EMDB) under accession codes EMD-50712 (ρ1-EM with THIP); EMD-50710 (ρ1-EM with CGP36742); EMD-50714 (ρ1-EM with racemic GABOB in a partially-locked state); and EMD-50713 (ρ1-EM with racemic GABOB in a desensitized state). The atomic coordinates have been deposited in the Protein Data Bank (PDB) under accession codes PDB-9FRE [https://doi.org/10.2210/pdb9FRE/pdb] (ρ1-EM with THIP); PDB-9FRB [https://doi.org/10.2210/pdb9FRB/pdb] (ρ1-EM with CGP36742); PDB-9FRH [https://doi.org/10.2210/pdb9FRH/pdb] (ρ1-EM with (R)-GABOB in a partially-locked state); PDB-9FRI [https://doi.org/10.2210/pdb9FRI/pdb] (ρ1-EM with (S)-GABOB in a partially-locked state); PDB-9FRF [https://doi.org/10.2210/pdb9FRF/pdb] (ρ1-EM with (R)-GABOB in a desensitized state); and PDB-9FRG [https://doi.org/10.2210/pdb9FRG/pdb] (ρ1-EM with (S)-GABOB in a desensitized state). Previously published structures referenced for comparison are available in the PDB under accession codes PDB-4MR8 [https://doi.org/10.2210/pdb4MR8/pdb]; PDB-7QND [https://doi.org/10.2210/pdb7QND/pdb]; PDB-8OP9 [https://doi.org/10.2210/pdb8OP9/pdb]; PDB-8OQ6 [https://doi.org/10.2210/pdb8OQ6/pdb]; PDB-8OQ7 [https://doi.org/10.2210/pdb8OQ7/pdb]; PDB-8RH7 [https://doi.org/10.2210/pdb8RH7/pdb]; and PDB-8RH8 [https://doi.org/10.2210/pdb8RH8/pdb]. Source data are provided with this paper.

## Code availability

MD simulation trajectories and parameter files, as well as code for calculating and plotting rmsds, are available on Zenodo [https://zenodo.org/records/15328258].

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

## Acknowledgements

We thank members of Molecular Biophysics Stockholm for feedback on the project and manuscript, and staff at the Swedish National Cryo-EM Facility for data collection and support. The data were collected at the Cryo-EM Swedish National Facility funded by the Knut and Alice Wallenberg, Family Erling Persson and Kempe Foundations, SciLifeLab and Stockholm University. MD simulations were performed using the computing facilities of the Swedish National Infrastructure for Computing (SNIC 2022/3–40), and supported by BioExcel (EuroHPC grant no. 101093290). J.C. was supported by an EMBO Postdoctoral Fellowship, C.F. by grant FV-5.1.2-0523-19 from Stockholm University, and R.J.H. and E.L. by grants from the Knut and Alice Wallenberg Foundation (2023.0254), the Swedish Research Council (2019-02433, 2021-05806) and the Swedish e-Science Research Centre.

## Author contributions

C.F. and J.C. performed the biochemistry, cryo-EM sample preparation and data processing. C.F. performed model building, refinement, structural analysis and MD simulations. J.C. performed electrophysiology. R.J.H. and E.L. supervised the project. All authors contributed to the manuscript writing and revision.

## Funding

## Competing interests

The authors declare no competing interests.
