## [Transparent Peer Review file · Nature Communications]

Cryo-EM structures of $\rho 1$ GABA_A receptors with antagonist and agonist drugs

Corresponding Author: Professor Erik Lindahl

Version 0:

Reviewer comments:

Reviewer #1

(Remarks to the Author)

This manuscript presents the cryo-em structure of rho-type GABA receptors in the presence of agonist and antagonist. The main significance of the work is the different binding poses that lead to receptor activation or inhibition with compounds that are have different functional effects at other GABA receptors (e.g. THIP as an agonist of GABAA receptors). This work appears to be carried out carefully and the results highly likely to be correct.

The recent advances in cryo-em have resulted in many GABAA and GABAC receptors resolved, so the overall structures being presented are not novel (e.g. Fan et al, Nat Comm, 2024). The novelty is restricted to the different interactions at the binding site.

I have some reservations about the interpretation of the results, where the structures appear to be shoehorned into conceptual theories, rather than carefully considered:

The discussion around the primed or pre-activated state is confusing. The "primed", "pre-activated" or "flip" state are essential to describing the concept of partial agonists. Agonists at the binding site that have a greater affinity for the primed state result in greater efficacy, as do positive modulators such as benzodiazepines (Gielen and Smart, 2012, Lape 2008). Both GABOB and GABA are full agonists, and thus would not be expected to display different primed structures. Further, the primed state is short lived and would be expected to rapidly convert to the open and desensitized states as seen in the cryo-em structures.

Based on this, defining the GABOB-bound closed state as "primed" does muddy the waters of these concepts. Is it really "primed" in the sense that it increases the likelihood of activating the receptor, or is it a bound state prior to loop capping that is caught because of the low affinity of GABOB for the receptor? This is further complicated by the reference to the structure in the presence of the steroid inhibitor (and this should reference Fan et al 2024 instead of reference 31). After all if this structure is similar to one with a bound inhibitor, how can it be a state that precedes activation? The authors do provide some ambiguity in the discussion but the reader will be left with the definition that any conformational change by an agonist represents a primed state.

The authors should consider re-writing the manuscript to make it clear what they feel this state represents and why. A good deal of the novelty of the work is the argument that this represents a "primed" state, but that claim should be treated with caution. Specifically, there is a much stronger and clearer argument to be made from their data that receptor activation requires a displacement of loop C of greater than 1.8Å, and falls into a range of 1.8-4.3Å. This is an important finding and will help define the activation energy required to activate the receptor via the native GABA-binding site, but is lost in the need to define the primed state.

As far as I could tell the manuscript was free of typos and grammatical errors and the figures were well presented.

Reviewer #2

(Remarks to the Author)

The manuscript from Chen Fan and colleagues provides an in-depth molecular investigation into the orthostatic site of the ρ -

type GABAA receptor using cryo-EM, electrophysiology, and molecular dynamics simulations. The GABAergic system plays important roles in human physiology by modulating cell excitability as well as via second messengers in response to the signaling molecule GABA. Given the diversity of GABA binding pockets across GABAA and GABAB receptor subtypes, understanding the molecular determinants of these ligand-binding pockets is of significant value. This knowledge can inform the design of ligands that selectively bind specific pockets and elicit defined biological effects. The manuscript serves as an elegant example of this approach in action. Specifically, the study investigates two antagonists (THIP and CGP36742) and one agonist (GABOB) of ρ -type GABAA receptors, using the human $\rho 1$ construct as a model system.

Major comments:

1. The cryo-EM structure revealed that THIP binds to the $\rho 1$ receptor in a distinct conformation compared to the $\alpha 4\beta 3\delta$ receptor. The sequence comparison (Fig S6) suggested several key residues of the $\rho 1$ receptor, some of which has been investigated by the authors experimentally. Residue Y268, which is conserved in ρ subunits but replaced by a serine or threonine in the corresponding positions of $\alpha 4/\beta 3/\delta$ subunits, may contribute to the differential pharmacology of THIP. How do the authors interpret the role of residue Y268 in this context?
2. Electrophysiology experiments demonstrated that 4 mM GABOB effectively opens the channel. The presence of desensitized channel from cryo-EM is consistent with this expectation. However, 86.7% of the particles was resolved to be in the primed state. Could the authors provide insight into which cryo-EM conditions might have contributed to this apparent discrepancy between the electrophysiological data and the structural findings?

Minor comments:

1. In Figure 1e, the CGP36742 molecule appears slightly off-scale in comparison to the other compounds.

Reviewer #3

(Remarks to the Author)

Review: Cryo-EM structures of $\rho 1$ GABAA receptors with antagonist and agonist drugs

The study aims to investigate the binding and conformational impact of a selection of ligands including THIP, CGP36742, and GABOB, on the $\rho 1$ -GABAA receptors. In this work, cryo-EM structures of $\rho 1$ -GABAA together with these ligands are solved, and the structures together with functional and computational efforts are used to characterize binding poses and their effects on the receptor.

The study found that the THIP ligand has a distinctive pose compared to other subtypes, such as $\alpha 4\beta 3\delta$ $\rho 1$ -GABAA receptors, explaining its inverse effects across subtypes. CGP36742 is found to bind similarly to TPMPA, suggesting a shared mechanism among phosphinic acid inhibitors. The GABOB ligand is found to have a similar binding pose as GABA but induces a mix of primed and desensitized conformation, and is concluded to likely be responsible for its weaker agonist activity. These findings provide mechanistic insights into how these drugs interact with $\rho 1$ -GABAA receptors and can contribute to advancing the understanding of the receptor's role in neurological conditions and guiding future drug development efforts.

Remarks to the author;

Major remarks

Page 6, paragraph 2: The authors superposed the resting-like THIP complex with a previous GABA-bound desensitized structure to visualize structural changes induced by antagonist versus agonist binding. The authors found that the bulky heterocycles of THIP were relatively protruded toward loop C, further concluding that this would obstruct its lockdown over the antagonist. However, ligand binding is a flexible process where both the pocket and the ligand adapt for optimal complementarity, hence superposition is not an optimal approach for comparing binding poses. I believe docking studies would be a more appropriate tool to evaluate the fit or clash of ligands in the binding pocket.

Page 9, Please clarify the statement claiming that "The CGP36742 pose from our $\rho 1$ -EM structure could be aligned into this GABAB receptor binding pocket on its equivalent aminopropyl and phosphinic acid moieties without clashing with any surrounding residues". Superposition is not a measurement for evaluating a compound's fit into a "new" pocket. When a compound is taken from one receptor pocket and placed into another receptor pocket, flexibility is necessary for adjusting the new structure and the pocket according to each other, for further evaluation of the complementarity.

Page 9, Last paragraph: Please clarify how the alignment of (S)-4ACBPBA in $\rho 1$ -EM and the GABA B receptor was performed. It seems the authors don't distinguish between the terms "alignment" and "superposition", and further how does the author encounter flexibility when enforcing/placing a new ligand into a binding pocket? If the authors do not encounter receptor and/or ligand flexibility, how do they know that the observed clashes (or absence of clashes) are not artificially introduced due to the lack of flexibility?

Figure 4: Please see the previous comment regarding using alignment for pose comparison.

Page 11-13 (MD simulation): Using ligand RMSD as the primary metric throughout the MD simulations of GABOB

enantiomers may not fully capture the mechanistic insights related to receptor activation. While the RMSD results indicate that both enantiomers exhibit some degree of movement, this observation alone does not provide sufficient evidence regarding their functional differences or activity (only leads to the conclusion that both of the enantiomers show some movement throughout the simulations). Including a negative control (an inactive compound) or positive control (like GABA) would perhaps provide more meaningful comparisons? Additionally, the small differences in RMSD (median below 2 Å) may not reflect significant structural changes. Since activation involves conformational changes in the receptor (such as those in loop C, as mentioned elsewhere in the manuscript), a focus on how the ligand induces these changes would offer deeper mechanistic insights. Therefore, incorporating other structural metrics or analyses that reflect the conformational dynamics of the receptor might provide a more comprehensive understanding of ligand-induced receptor activation.

Minor remarks;

In the abstract, the statement “.. structures, recordings and simulations to characterize the binding and conformational impact of the drugs .. on a human ρ 1-GABAA receptor” is lacking clarity: Which structures are referred to (newly solved Cryo-EM presented here, modeled structures, etc?) and what kind of recordings?

Page 2, last paragraph: Could the authors please provide a reference for the statement "In humans, GABAA receptors are homo- or hetero-pentamers formed from a selection of 19 different subunits (α 1-6, β 1-3, γ 1-3, ρ 1-3, δ , ϵ , π and θ)."

Page 3, Section 2: the statement “Several drugs act on ρ -type GABAA” can give an incorrect impression that there are several approved drugs targeting and binding to ρ -type GABAA. I would recommend a more neutral word like binder or ligand.

Page 3, Section 3: It is not clear which structures the authors are referring to in the first sentence of this section “combining structures, electrophysiology, and molecular dynamics (MD) simulations..” Solved 3D structures, Modeled structures, etc.? The dynamics/conformational states of the receptor have not been described in the introduction and the states mentioned in this section could for clarity be explained (primed and desensitized states).

Page 7, Figure 2c-d) For clarity, the reviewer suggests keeping the color codes consistent and using the color “Tan” for THIP as previously. Also, it would be preferable to visualize the ligand with polar hydrogens and bond type as the current depiction does not allow a full understanding of the protein-ligand interactions.

Page 8: Please clarify what is meant by the term “nearly” superimposable with your previous TPMPA structure. Which methods were applied to compare: superposition followed by rmsd? If yes, how was this done, and which threshold was used for justifying the term “nearly superimposable”

Page 9, First paragraph: “Given the potency of”.. The reviewer finds this term a bit misleading and believes that the right phrasing should be something like “Given the ability of..”

Page 11, Figure 5: The RMSD of a specific amino acid (Ser264) is used as a metric for differentiating receptor states/conformations, but as this is not mentioned in the figure legend, the informative value of the right half of the figure is lacking. Also, it would be beneficial to show the stick view (including polar hydrogens).

Fig.6 a) and b) Modeling is a broad term including multiple approaches and a more specific term to describe how the ligands were placed into the pocket would be recommended. c) and d) The wording “zoomed view” could be omitted for conciseness e.g. protein-ligand interaction diagram?

Fig.7c) Orthosteric-site expansion/compaction-based ligand identity might be rightly claimed as well as the size of the ligand, however, all agonists/antagonists in the figure are in the same size range and would all be considered almost fragment-like.

In the discussion, the cross-reactivity of CGP36742 to GABAB is explained. However, many of the known compounds showing activity towards the GABA B receptor contain phosphinic acid moieties. Could this shared mechanism of action among phosphinic acid inhibitors be a problem for receptor selectivity?

Molecular dynamics simulations: Did the authors try to model the missing residues in the M3-M4 loop? When excluded, this could potentially affect protein dynamics, especially in the TMD (transmembrane domain), depending on their length and relevance. Could the author comment on this?

CGenFF is commonly used to generate parameters for ligands, but as the parameterization is often automated, did the authors manually verify that the generated parameters (charges, dihedrals, etc.) for GABOB were accurate to avoid unusual bond angles or charges that could affect the dynamics?

Reviewer #4

(Remarks to the Author)

Version 1:

Reviewer comments:

Reviewer #1

(Remarks to the Author)

The authors have fully addressed my concerns, and it is a credit to them the way they have re-written the document with clear and specific language that distinguishes between different concepts.

Reviewer #2

(Remarks to the Author)

I thank the authors for their thorough response to all the comments! I have no further questions.

Reviewer #3

(Remarks to the Author)

Response to Revised Manuscript

I would like to thank the authors for their thorough responses to the comments raised in my initial review. A considerable effort has gone into addressing each point, and I truly appreciate the additional analyses, clarifications, and revisions that have been incorporated into the revised manuscript.

I find that my comments have been addressed satisfactorily, and I appreciate the authors' openness to the feedback. The revised manuscript represents meaningful improvement, and I have no further major concerns.

Thank you again for your careful attention to the review.

Best regards,

Dr. Evenseth

Reviewer #4

(Remarks to the Author)

Response to Referees:

Cryo-EM structures of p1 GABA_A receptors with antagonist and agonist drugs

Reviewer #1 (Remarks to the Author):

This manuscript presents the cryo-em structure of rho-type GABA receptors in the presence of agonist and antagonist. The main significance of the work is the different binding poses that lead to receptor activation or inhibition with compounds that have different functional effects at other GABA receptors (e.g. THIP as an agonist of GABA_A receptors). This work appears to be carried out carefully and the results highly likely to be correct.

The recent advances in cryo-em have resulted in many GABA_A and GABA_C receptors resolved, so the overall structures being presented are not novel (e.g. Fan et al, Nat Comm, 2024). The novelty is restricted to the different interactions at the binding site.

I have some reservations about the interpretation of the results, where the structures appear to be shoehorned into conceptual theories, rather than carefully considered:

1: The discussion around the primed or pre-activated state is confusing. The "primed", "pre-activated" or "flip" state are essential to describing the concept of partial agonists. Agonists at the binding site that have a greater affinity for the primed state result in greater efficacy, as do positive modulators such as benzodiazepines (Gielen and Smart, 2012, Lape 2008). Both GABOB and GABA are full agonists, and thus would not be expected to display different primed structures. Further, the primed state is short lived and would be expected to rapidly convert to the open and desensitized states as seen in the cryo-em structures.

Based on this, defining the GABOB-bound closed state as "primed" does muddy the waters of these concepts. Is it really "primed" in the sense that it increases the likelihood of activating the receptor, or is it a bound state prior to loop capping that is caught because of the low affinity of GABOB for the receptor? This is further complicated by the reference to the structure in the presence of the steroid inhibitor (and this should reference Fan et al 2024 instead of reference 31). After all if this structure is similar to one with a bound inhibitor, how can it be a state that precedes activation? The authors do provide some ambiguity in the discussion but the reader will be left with the definition that any conformational change by an agonist represents a primed state.

The authors should consider re-writing the manuscript to make it clear what they feel this state represents and why. A good deal of the novelty of the work is the argument that this represents a "primed" state, but that claim should be treated with caution. Specifically, there is a much stronger and clearer argument to be made from their data that receptor activation requires a displacement of loop C of greater than 1.8Å, and falls into a range of 1.8-4.3Å. This is an important finding and will help define the activation energy required to activate the receptor via the native GABA-binding site, but is lost in the need to define the primed state.

As far as I could tell the manuscript was free of typos and grammatical errors and the figures were well presented.

We thank the reviewer for noting the generally careful nature of this work. We regret all the more then, muddying the important and precise definition of the *primed* state, by ascribing it to the predominant GABOB-bound class—an issue also raised in Comments 2.2 and 3.9. In retrospect, our election to annotate this structure as *primed* was indeed biased by its similarity to the structure we previously reported with GABA plus the inhibitor estradiol. In that work, we extrapolated that the inhibitor had trapped a subpopulation of receptors in a partially active intermediate, potentially *primed* state. Both the GABOB and GABA+estradiol complexes can be more rigorously described in structural terms, with loop C only *partially locked* over the agonist site, such that the allosteric signal fails to propagate to pore opening (Fig.5).

Of course, this partially-locked state is trapped by distinct influences in the presence of GABOB versus GABA+estradiol. In the latter case, binding of estradiol to an allosteric site appears to suppress coupling of GABA binding to ECD rotation. In the former, partial occupancy of GABOB—consistent with its low potency and slow kinetics relative to the canonical agonist GABA—may be insufficient to stimulate complete rotation/activation of the ECD in a subset of p1 particles. Although precise occupancy is difficult to discern in the context of cryo-EM signal averaging, it is interesting that the desensitized state in the same GABOB dataset exhibited relatively higher local resolution in the ECD (Supplementary Fig.4).

On further consideration, we have substituted the term *partially locked* for *primed* throughout the text and figures, and clarified the comparison to the inhibitor-bound structure in Results and Discussion, including the correction to Reference 31.

Results (p. 4 l. 111): **In another class (87% of resolved particles), the ECD was not fully activated, with loop C only partially locked over the agonist site; the pore remained at rest. This structure, corresponding to neither resting nor desensitized states, aligned well with a previous structure determined with the negative modulator 17 β -estradiol (E2)³¹ (Supplementary Fig.4e,f) (Table 2). Notably, E2 binds at the ECD-TMD interface, where it appears to disrupt allosteric transitions induced by GABA binding, including lockdown of loop C over the agonist site. In contrast, we observed no ligand density at the equivalent interface in GABOB-bound structures, indicating a distinct mechanism of stabilizing the partially-locked state.**

Results (p. 12 l. 250): **Interestingly, although all structures in this work contained ligands in the extracellular orthosteric site, local resolution in the ECD was relatively higher in the GABOB-desensitized state; in resting and partially-locked structures, resolution was similar between the domains (Supplementary Fig.4a-d), suggesting that channel activation is associated with stabilization of the ECD and mobilization of the TMD.**

Discussion (p. 14 l. 321): **Antagonists such as THIP and phosphinic acids clearly obstruct lockdown of loop C altogether, resulting in an expansive orthosteric site superimposable with that of the apo structure (Fig.7).**

Discussion (p. 15 l. 337): At the other extreme, agonists such as GABA enable substantial lockdown of loop C, resulting in compaction of the orthosteric site and rotation of the ECD relative to the TMD (Fig.7a,b). Alongside this apparent activated-desensitized state, treatment with the weaker agonist GABOB promoted a subclass exhibiting only partial lockdown of loop C relative to the apo form (Fig.7a,c). We previously reported a similar partially-locked structure in the presence of the inhibitor E2, which might allosterically trap the so-called primed state, or some metastable intermediate on the pathway from resting to open³¹. In the absence of allosteric inhibition, GABOB appears to favor a partially-locked population by different means.

Reviewer #2 (Remarks to the Author):

The manuscript from Chen Fan and colleagues provides an in-depth molecular investigation into the orthosteric site of the ρ -type GABAA receptor using cryo-EM, electrophysiology, and molecular dynamics simulations. The GABAergic system plays important roles in human physiology by modulating cell excitability as well as via second messengers in response to the signaling molecule GABA. Given the diversity of GABA binding pockets across GABAA and GABAB receptor subtypes, understanding the molecular determinants of these ligand-binding pockets is of significant value. This knowledge can inform the design of ligands that selectively bind specific pockets and elicit defined biological effects. The manuscript serves as an elegant example of this approach in action. Specifically, the study investigates two antagonists (THIP and CGP36742) and one agonist (GABOB) of ρ -type GABAA receptors, using the human $\rho 1$ construct as a model system.

Major comments:

2.1: The cryo-EM structure revealed that THIP binds to the $\rho 1$ receptor in a distinct conformation compared to the $\alpha 4\beta 3\delta$ receptor. The sequence comparison (Fig S6) suggested several key residues of the $\rho 1$ receptor, some of which has been investigated by the authors experimentally. Residue Y268, which is conserved in ρ subunits but replaced by a serine or threonine in the corresponding positions of $\alpha 4/\beta 3/\delta$ subunits, may contribute to the differential pharmacology of THIP. How do the authors interpret the role of residue Y268 in this context?

We thank the reviewer for their positive, and are all the more grateful for catching the error in Supplementary Fig.6. In our original manuscript, we mistakenly placed a blue marker on Y272, a residue that is indeed substituted by S or T in alternative GABA_AR subunits. However, Y272 is not in fact a close contact to THIP; that marker should have been placed on Y262, already shown in Fig.2. As seen in the annotated clip below, the aromatic-box residue Y268 is highly conserved across subtypes, varying only to F in δ subunits; this position seems unlikely to contribute to specificity.

We have corrected Supplementary Fig.6 accordingly.

2.2: Electrophysiology experiments demonstrated that 4 mM GABOB effectively opens the channel. The presence of desensitized channel from cryo-EM is consistent with this expectation. However, 86.7% of the particles was resolved to be in the primed state. Could the authors provide insight into which cryo-EM conditions might have contributed to this apparent discrepancy between the electrophysiological data and the structural findings?

Building on modifications in response to Reviewer 1, we have expanded the text to address this important point more clearly.

Discussion (p. 15 l. 344): **Given its low potency and slow kinetics relative to GABA, GABOB occupancy may be insufficient to stimulate complete rotation/activation of the ECD in a subset of p1 particles. It is also plausible that non-physiological sample conditions, such as embedding in a lipid nanodisc or non-instantaneous plunge-freezing, may relatively destabilize the GABOB-activated state. Although agonist binding to at least 3 of 5 subunit interfaces is thought to enable p1 activation⁴³, the limited diffusive volume and high local receptor density on the cryo-EM grid may limit ligand accessibility, even in the presence of a supersaturating bulk agonist concentration. Indeed, it remains unclear why a fully activated-open structure of p1 remains experimentally inaccessible with either GABA or GABOB^{27,31}.**

2.3: Minor comment: In Figure 1e, the CGP36742 molecule appears slightly off-scale in comparison to the other compounds.

We have updated **Fig.1e** such that CGP36742 is scaled to match the other ligands.

Reviewer #3 (Remarks to the Author):

Review: Cryo-EM structures of p1 GABA_A receptors with antagonist and agonist drugs

The study aims to investigate the binding and conformational impact of a selection of ligands including THIP, CGP36742, and GABOB, on the p1-GABA_A receptors. In this work, cryo-EM structures of p1-GABA_A together with these ligands are solved, and the structures together with

functional and computational efforts are used to characterize binding poses and their effects on the receptor.

The study found that the THIP ligand has a distinctive pose compared to other subtypes, such as $\alpha 4\beta 3\delta$ $\rho 1$ -GABAA receptors, explaining its inverse effects across subtypes. CGP36742 is found to bind similarly to TPMPA, suggesting a shared mechanism among phosphinic acid inhibitors. The GABOB ligand is found to have a similar binding pose as GABA but induces a mix of primed and desensitized conformation, and is concluded to likely be responsible for its weaker agonist activity. These findings provide mechanistic insights into how these drugs interact with $\rho 1$ -GABAA receptors and can contribute to advancing the understanding of the receptor's role in neurological conditions and guiding future drug development efforts.

Remarks to the author;

Major remarks

3.1: Page 6, paragraph 2: The authors superposed the resting-like THIP complex with a previous GABA-bound desensitized structure to visualize structural changes induced by antagonist versus agonist binding. The authors found that the bulky heterocycles of THIP were relatively protruded toward loop C, further concluding that this would obstruct its lockdown over the antagonist. However, ligand binding is a flexible process where both the pocket and the ligand adapt for optimal complementarity, hence superposition is not an optimal approach for comparing binding poses. I believe docking studies would be a more appropriate tool to evaluate the fit or clash of ligands in the binding pocket.

We are grateful for the reviewer's recognition of the potential mechanistic insights from this work, and particularly for this thoughtful point. Indeed, computational docking could allow far better sampling of plausible ligand-protein interactions, as well as quantitative binding scores, than the superpositions we initially applied. We have now investigated THIP binding to resting-like and desensitized structures using AutoDock Vina, resulting in the first two panels of new **Supplementary Fig.7**. Consistent with our superposition-based observations, THIP docks more favorably to the resting-like than to the desensitized state. We have also added relevant text to Results and Methods.

Results (p. 6 l. 137): **Superposing the resting-like THIP complex with our previous GABA-bound desensitized structure²⁷ (Fig.2c,d) highlighted structural changes induced by antagonist versus agonist binding. The amino and hydroxyl groups of THIP roughly overlapped those of GABA; however, the bulky heterocycles of THIP were relatively protruded toward loop C, obstructing its lockdown over the antagonist. Moreover, computational docking of THIP into these two $\rho 1$ structures produced more favorable binding energy scores in the resting-like versus desensitized state (Supplementary Fig.7a,b).**

Methods (p. 18 l. 435): **Docking was performed using Autodock Vina⁵⁶, with search volume $15 \text{ \AA} * 15 \text{ \AA} * 15 \text{ \AA}$ around the center of each pocket.**

3.2: Page 9, Please clarify the statement claiming that “The CGP36742 pose from our p1-EM structure could be aligned into this GABAB receptor binding pocket on its equivalent aminopropyl and phosphinic acid moieties without clashing with any surrounding residues”. Superposition is not a measurement for evaluating a compound’s fit into a “new” pocket. When a compound is taken from one receptor pocket and placed into another receptor pocket, flexibility is necessary for adjusting the new structure and the pocket according to each other, for further evaluation of the complementarity.

Again, we are grateful for this admonition, and have applied the same computational docking approach described in response to the previous comment to verify the binding pose of CGP36742, resulting in new Supplementary Fig.7c. Reassuringly, the best-scoring pose is comparable to the superposition previously shown in Fig.4b. We have also clarified references to ligands that could be “aligned,” and added relevant text to Results and Methods.

Results (p. 9 l. 201): The CGP36742 pose from our p1-EM structure could be placed in this pocket in the GABA_B receptor complex by superposition on its equivalent aminopropyl and phosphinic acid moieties (Fig.4b), or by computational docking (Supplementary Fig.7c), with similar poses showing no evident clashes and a favorable binding energy score.

3.3: Page 9, Last paragraph: Please clarify how the alignment of (S)-4ACPBPA in p1-EM and the GABA B receptor was performed. It seems the authors don’t distinguish between the terms “alignment” and “superposition”, and further how does the author encounter flexibility when enforcing/placing a new ligand into a binding pocket? If the authors do not encounter receptor and/or ligand flexibility, how do they know that the observed clashes (or absence of clashes) are not artificially introduced due to the lack of flexibility?

Figure 4: Please see the previous comment regarding using alignment for pose comparison.

We have now clarified the Results to distinguish protein alignment from ligand superposition or docking, and the Methods to clarify our use of the “align” command in UCSF Chimera to carry out superposition.

Results (p. 9 l. 210): Superposition of (S)-4-ACPBPA with CGP36742 in our p1-EM complex showed this ligand could be accommodated without modification (Fig.4e).

Methods (p. 18 l. 424): For initial ligand comparisons across receptor families, ligands (CGP36742, (S)-4-ACPBPA) were superposed using the "align" command in UCSF Chimera⁵⁴ to match corresponding non-hydrogen atoms in the target complex (CGP35348, CGP36742).

Regarding flexibility effects in the GABA_B receptor (green), the illustration at right depicts several alternative orientations of the W65 side chain (gray). All likely rotamers closely approach or directly clash with the superposed (S)-4ACPBPA (tan), suggesting that side-chain flexibility alone would not accommodate binding of this ligand. We also applied computational docking to model (S)-4ACPBPA in both p1-EM and the GABA_B receptor, resulting in new **Supplementary Fig.7d-e**. Consistent with our previous, more rigid approximations (Fig.4e-f), the ligand docked more favorably in p1 than in the GABA_B receptor. We now describe this observation in Results.

Results (p. 9 l. 211): **Conversely, superposing this compound with CGP35348 in the GABA_B receptor resulted in a clash of the amino group with residue W65, suggesting a molecular basis for receptor specificity (Fig.4f). Consistent with these predictions, computational docking of (S)-4ACPBPA produced more favorable binding energy scores in p1-EM than in the GABA_B receptor (Supplementary Fig.7d,e).**

3.4: Page 11-13 (MD simulation): Using ligand RMSD as the primary metric throughout the MD simulations of GABOB enantiomers may not fully capture the mechanistic insights related to receptor activation. While the RMSD results indicate that both enantiomers exhibit some degree of movement, this observation alone does not provide sufficient evidence regarding their functional differences or activity (only leads to the conclusion that both of the enantiomers show some movement throughout the simulations). Including a negative control (an inactive compound) or positive control (like GABA) would perhaps provide more meaningful comparisons? Additionally, the small differences in RMSD (median below 2 Å) may not reflect significant structural changes. Since activation involves conformational changes in the receptor (such as those in loop C, as mentioned elsewhere in the manuscript), a focus on how the ligand induces these changes would offer deeper mechanistic insights. Therefore, incorporating other structural metrics or analyses that reflect the conformational dynamics of the receptor might provide a more comprehensive understanding of ligand-induced receptor activation.

We appreciate the opportunity to explore GABOB dynamics in more detail, particularly with respect to protein structural metrics. To this end, we investigated the hydrogen bond between Y268 and R179(-), which we found in previous p1 studies to clearly distinguish agonist-bound from -unbound systems. Either of the GABOB enantiomers appeared to sustain Y268-R179 interactions more tightly than in the resting structure, similar to our previous simulations with GABA. We now document these analyses in two new figure panels (**Fig.6h,i**), and describe them in Results.

Results (p. 12 l. 273): **Previously we have also observed an intersubunit hydrogen bond between residue Y268 on the principal loop C and R179(-) on the complementary loop F, which**

characterizes ligand-bound versus -unbound states²⁷ (Fig.6h). Similar to our previous simulations with GABA, either of the GABOB enantiomers sustained tighter Y268-R179 interactions than in the resting structure, though a modest trend towards even tighter contacts in the presence of (R)-GABOB appeared consistent with the slightly greater potency of this enantiomer (Fig.6i). Interestingly, this interaction was retained in the partially-locked as well as desensitized systems, despite the limited extent of ECD activation.

Minor remarks

3.5: In the abstract, the statement “.. structures, recordings and simulations to characterize the binding and conformational impact of the drugs .. on a human ρ 1-GABAA receptor” is lacking clarity: Which structures are referred to (newly solved Cryo-EM presented here, modeled structures, etc?) and what kind of recordings?

We have now clarified the Abstract to specify the type of data reported.

Abstract (p. 2 l. 21): Here we report four cryo-EM structures with previously unresolved ligands, electrophysiology recordings, and molecular dynamics simulations to characterize the binding and conformational impact of the drugs THIP (a non-opioid analgesic), CGP36742 (a phosphinic acid inhibitor), and GABOB (an anticonvulsant) on a human ρ 1 GABA_A receptor.

3.6: Page 2, last paragraph: Could the authors please provide a reference for the statement "In humans, GABAA receptors are homo- or hetero-pentamers formed from a selection of 19 different subunits (α 1-6, β 1-3, γ 1-3, ρ 1-3, δ , ϵ , π and θ)."

We have added Reference 4 to support the statement.

Reference 4: Olsen, R. W. & Sieghart, W. GABA_A receptors: subtypes provide diversity of function and pharmacology. *Neuropharmacology* 56, 141–148 (2009).

3.7: Page 3, Section 2: the statement “Several drugs act on ρ -type GABAA” can give an incorrect impression that there are several approved drugs targeting and binding to ρ -type GABAA. I would recommend a more neutral word like binder or ligand.

We have modified this sentence as suggested.

Introduction (p. 3 l. 67): Several ligands act on ρ -type GABA_A receptors...

3.8: Page 3, Section 3: It is not clear which structures the authors are referring to in the first sentence of this section “combining structures, electrophysiology, and molecular dynamics (MD) simulations..” Solved 3D structures, Modeled structures, etc.?

We have clarified the final paragraph of Introduction as suggested.

Introduction (p. 3 l. 82): Here, combining four original cryo-EM structures, electrophysiology and molecular dynamics (MD) simulations, we characterize the binding and structural impact of THIP, CGP36742, and GABOB on human $\rho 1$ GABA_A receptors.

3.9: The dynamics/conformational states of the receptor have not been described in the introduction and the states mentioned in this section could for clarity be explained (primed and desensitized states).

We regret this omission, and have expanded the Introduction to better contextualize pLGIC conformational cycling.

Introduction (p. 2 l. 44): In a generalized gating mechanism for pLGICs, binding of agonist to the unliganded resting state induces lockdown of loop C over the orthosteric site, and a long-range conformational transition spanning most of the protein². This transition is thought to progress through a primed state with increased ligand affinity but transient kinetics, an open state with an expanded hydrophobic gate at the midpoint across the membrane, and a desensitized state with a contracted gate at the intracellular end of the pore. Although open states of common GABA_A-receptor subtypes have proved difficult to resolve, a growing catalog of structures representing resting, desensitized, and intermediate states offer critical insights into conformational cycling and ligand modulation, as outlined below.

3.10: Page 7, Figure 2c-d) For clarity, the reviewer suggests keeping the color codes consistent and using the color "Tan" for THIP as previously. Also, it would be preferable to visualize the ligand with polar hydrogens and bond type as the current depiction does not allow a full understanding of the protein-ligand interactions.

With thanks for these corrections, colors and legends in Fig.2 have been updated. We have also added dashed lines to depict potential hydrogen bonds in Fig.2a. We tried also depicting polar hydrogens (see example at right), but our test readers found these representations cluttered and difficult to parse; we hope the reviewer will agree to the compromise.

3.11: Page 8: Please clarify what is meant by the term “nearly” superimposable with your previous TPMPA structure. Which methods were applied to compare: superposition followed by rmsd? If yes, how was this done, and which threshold was used for justifying the term “nearly superimposable”

We have clarified the Methods to define the tool used to superpose and calculate RMSD, and included those values in Results.

Results (p. 8 l. 180): **The complex was comparable to the apo and THIP structures (Table 2, Fig.3c), and superimposable with our previous structure with TPMPA (C α RMSD 0.035 Å (Fig.3d)...**

Methods (p. 18 l. 433): **RMSDs were calculated by aligning C α atoms in two given structures using the “match” command in UCSF Chimera.**

3.12: Page 9, First paragraph: “Given the potency of”.. The reviewer finds this term a bit misleading and believes that the right phrasing should be something like “Given the ability of..”

We have rephrased this sentence of Results as suggested.

Results (p. 9 l. 197): **Given the ability of CGP36742 to inhibit GABA_B as well as ρ 1 GABA_A receptors^{29,37}...**

3.13: Page 11, Figure 5: The RMSD of a specific amino acid (Ser264) is used as a metric for differentiating receptor states/conformations, but as this is not mentioned in the figure legend, the informative value of the right half of the figure is lacking. Also, it would be beneficial to show the stick view (including polar hydrogens).

We are grateful for the reviewer catching this omission, and have modified the legend to **Fig.5** accordingly. We have also edited the panels to show the S264 side chain, including the polar hydrogen as sticks.

3.14: Fig.6 a) and b) Modeling is a broad term including multiple approaches and a more specific term to describe how the ligands were placed into the pocket would be recommended.

We have substituted “refinement” for “modeling” in the legend to **Fig.6** as suggested.

3.15: c) and d) The wording “zoomed view” could be omitted for conciseness e.g. protein-ligand interaction diagram?

We have substituted “protein-ligand interactions” for “zoomed view” in the legend to **Fig.6** as suggested.

3.16: Fig.7c) Orthosteric-site expansion/compaction-based ligand identity might be rightly claimed as well as the size of the ligand, however, all agonists/antagonists in the figure are in the same size range and would all be considered almost fragment-like.

To avoid misunderstanding, we have modified the legend to Fig.7 to indicate “antagonists to strong agonists.”

3.17: In the discussion, the cross-reactivity of CGP36742 to GABAB is explained. However, many of the known compounds showing activity towards the GABA B receptor contain phosphinic acid moieties. Could this shared mechanism of action among phosphinic acid inhibitors be a problem for receptor selectivity?

We have modified the Discussion to emphasize this important point.

Discussion (p. 14 l. 323): **Despite the shared binding profile among phosphinic acid inhibitors in both GABA-receptor families, modification of the aminopropyl tail in (S)-4-ACPBPA, or of the butyl tail in CGP35348, successfully confers preference for GABA_A and GABA_B receptors respectively, indicating that selective modulators can be engineered on this scaffold.**

3.18: Molecular dynamics simulations: Did the authors try to model the missing residues in the M3-M4 loop? When excluded, this could potentially affect protein dynamics, especially in the TMD (transmembrane domain), depending on their length and relevance. Could the author comment on this?

We have expanded the Methods to clarify this point.

Methods (p. 19 l. 459): **This approach carries the risk of enabling non-physiological dynamics around deletion endpoints; on the other hand, it avoids introducing information without direct experimental evidence, for example by modeling the missing loop, inserting a shorter sequence, or applying positional restraints. Given that the sub-microsecond timescales of our simulations are not expected to sample allosteric effects of the intracellular domain on drug-binding sites, which are located at least 70 Å away on the opposite side of the membrane, and that overall Cα RMSDs remained within 4 Å (Supplementary Fig.9), this approach appeared to be a reasonable and parsimonious compromise.**

3.19: CGenFF is commonly used to generate parameters for ligands, but as the parameterization is often automated, did the authors manually verify that the generated parameters (charges, dihedrals, etc.) for GABOB were accurate to avoid unusual bond angles or charges that could affect the dynamics?

With regrets for not making this clear, visual and quantitative inspection of both GABOB ligands indicated reasonable parameterization. For reviewer reference, closer views of both enantiomers are illustrated below, with explicit hydrogens (white) and otherwise colored by heteroatom (nitrogen, blue; oxygen, red).

(R)-GABOB

(S)-GABOB

We have included the parameter files in the Zenodo deposition associated with this work (<https://zenodo.org/records/14642302>), and updated the Methods to comment specifically on the penalty scores associated with the parameters generated for both GABOB enantiomers.

Methods (p. 20 l. 472): **Visual and quantitative inspection of the resulting parameters (param penalty 4.600, charge penalty 10.008) indicated reasonable description of both ligands.**

Reviewer #4 (Remarks to the Author):

4: I co-reviewed this manuscript with one of the reviewers who provided the listed reports. This is part of the Nature Communications initiative to facilitate training in peer review and to provide appropriate recognition for Early Career Researchers who co-review manuscripts.

We thank the reviewer for contributing to this helpful feedback.